# Discovering Latent Causal Graphs from Spatiotemporal Data

**Kun Wang** [* 1]  **Sumanth Varambally** [* 2]  **Duncan Watson-Parris** [2 3]  **Yi-An Ma** [2 1]  **Rose Yu** [1 2]

## Abstract

Many important phenomena in scientific fields like climate, neuroscience, and epidemiology are naturally represented as spatiotemporal gridded data with complex interactions. Inferring causal relationships from these data is a challenging problem compounded by the high dimensionality of such data and the correlations between spatially proximate points. We present SPACY (SPAtiotemporal Causal discoverY), a novel framework based on variational inference, designed to model latent time series and their causal relationships from spatiotemporal data. SPACY alleviates the high-dimensional challenge by discovering causal structures in the latent space. To aggregate spatially proximate, correlated grid points, we use spatial factors, parametrized by spatial kernel functions, to map observational time series to latent representations. Theoretically, we generalize the problem to a continuous spatial domain and establish identifiability when the observations arise from a nonlinear, invertible function of the product of latent series and spatial factors. Using this approach, we avoid assumptions that are often unverifiable, including those about instantaneous effects or sufficient variability. Empirically, SPACY outperforms state-of-the-art baselines on synthetic data, even in challenging settings where existing methods struggle, while remaining scalable for large grids. SPACY also identifies key known phenomena from real-world climate data. An implementation of SPACY is available at https://github.com/Rose-STL-Lab/SPACY/

---

[*]Equal contribution [1]Department of Computer Science and Engineering, University of California, San Diego, La Jolla, USA [2]Halıcıoğlu Data Science Institute, University of California, San Diego, La Jolla, USA [3]Scripps Institution of Oceanography, University of California, San Diego, La Jolla, USA. Correspondence to: Sumanth Varambally <svarambally@ucsd.edu>, Kun Wang <bw5889@princeton.edu>.

*Proceedings of the 42ⁿᵈ International Conference on Machine Learning*, Vancouver, Canada. PMLR 267, 2025. Copyright 2025 by the author(s).

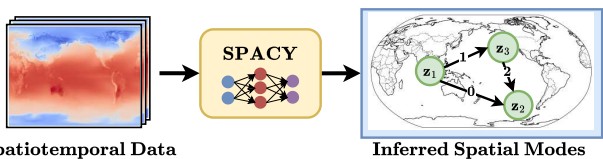

**Spatiotemporal Data**                    **Inferred Spatial Modes and Latent Causal Graph**

Figure 1: SPACY jointly infers latent time series and the underlying causal graph from gridded time-series data by identifying spatial modes of variability.

## 1. Introduction

In many scientific domains such as climate science, neurology, and epidemiology, low-level sensor measurements generate high-dimensional observational data. These data are naturally represented as gridded time series, with interactions that evolve over both space and time. Discovering causal relationships from spatiotemporal data is of great scientific importance. It allows researchers to predict future states, intervene in harmful trends, and develop new insights into the underlying mechanisms. For example, in climate science, the study of teleconnections (Liu et al., 2023), the interactions between regions thousands of kilometers away, is important to understanding how climate events in one part of the world may affect weather patterns in distant locations.

Despite the plethora of work on causal structure learning from time series data (Granger, 1969; Hyvärinen et al., 2010; Runge, 2020a; Tank et al., 2021; Gong et al., 2023; Cheng et al., 2023), they face significant challenges for high-dimensional spatiotemporal data. A primary reason is scalability. The high dimensionality of gridded data makes it difficult for many of these techniques, especially those relying on conditional independence tests, to scale effectively (Glymour et al., 2019). Additionally, spatially proximate points often exhibit redundant and highly correlated time series. Conditioning on nearby correlated points can obscure true causal relationships between distant locations, reducing the statistical power of conditional independence tests and leading to inaccurate results (Tibau et al., 2022).

Recent advances in spatiotemporal causal discovery typically follow a two-stage process: first, dimensionality reduction is applied to extract a small number of latent time series from the original grid of time series; then, causal discovery is performed on these low-dimensional representa-

| Methods | Non-linear SCM | Non-linear mapping from latents to observations | Instantaneous Links | Allows multiple latent parents |
|---|---|---|---|---|
| Mapped-PCMCI (Tibau et al., 2022) | ✗ | ✗ | ✗ | ✗ |
| Linear-Response (Falasca et al., 2024) | ✗ | ✗ | ✗ | ✗ |
| LEAP (Yao et al., 2022b) | ✓ | ✓ | ✗ | ✓ |
| TDRL (Zhao et al., 2023a) | ✓ | ✓ | ✗ | ✓ |
| CDSD (Brouillard et al., 2024) | ✓ | ✓ | ✗ | ✗ |
| **SPACY (ours)** | ✓ | ✓* | ✓ | ✓ |

Table 1: Assumptions of various spatiotemporal causal discovery algorithms. SPACY supports a non-linear mapping from latent variables to observations via an invertible transformation of the product between latent time series and spatial factors (denoted by $*$).

tions. Examples of this approach include Tibau et al. (2022) and Falasca et al. (2024). However, these two-stage approaches perform dimensionality reduction independent of the causal structure, potentially leading to latent representations that obscure the relationships among causally relevant entities. Another important line of research is causal representation learning from time series data (Schölkopf et al., 2021). While approaches like Yao et al. (2022b;a); Chen et al. (2024) infer latent time series from high-dimensional data, they do not incorporate spatial priors, making them less suitable for spatiotemporal causal discovery. Brouillard et al. (2024) learn to map each time series to a latent variable under the single-parent assumption, that is, each observed variable is influenced by only one latent variable. However, this assumption can be overly restrictive in spatiotemporal systems where observed variables are often influenced by multiple interacting latent factors (e.g., atmospheric patterns, ocean currents) that jointly drive their behavior.

We present SPAtiotemporal Causal DiscoverY (SPACY), a novel causal representation learning framework for spatiaotemporal data, to address these limitations (Figure 1). To model spatial variability on the grid, we introduce spatial factors, parametrized by spatial kernel functions such as Radial Basis Functions (RBFs). These spatial factors aggregate proximate grid points and map the observed variables to their corresponding latent time series. Since SPACY performs causal discovery on these lower-dimensional representations, SPACY is naturally scalable. We derive an evidence lower bound to learn the spatial factors, latent time series as well as the causal graph. Our approach jointly infers both the latent time series and the underlying causal graph in an *end-to-end* manner.

We also prove the identifiability of our framework for a continuous spatial domain with an infinite resolution. We show that when the observations are a nonlinear, invertible function of the product of latent series and spatial factors, we can leverage the overdetermined structure of the system to recover spatial factors and latent time series (up to permutation and scaling). Notably, compared to previous works,

our framework can handle instantaneous edges while also allowing observed variables to be associated with multiple latent parents.

Our main contributions can be summarized as follows.

1. We introduce SPACY, a novel variational inference-based spatiotemporal causal discovery framework that jointly infers latent time series and the underlying causal graph.
2. Theoretically, we analyze the identifiability of our system for continuous spatial domains. We show that the latent factors are identifiable up to permutation and scaling under both linear and nonlinear invertible mappings between the latent and observed variables.
3. Experimentally, we demonstrate the strong performance of our method on both synthetic and real-world datasets. SPACY accurately recovers causal links in scenarios involving both linear and nonlinear interactions, including settings where existing methods fail. SPACY is also highly scalable, and can infer causal links from high-dimensional grids (upto $250 \times 250$).

## 2. Related Work

**Causal Discovery from Time Series data.** A prominent approach to time series causal discovery is based on Granger causality (Granger, 1969), and its extension using neural networks (Khanna & Tan, 2020; Tank et al., 2021; Löwe et al., 2022; Cheng et al., 2023; 2024). However, Granger causality only captures predictive relationships and ignores instantaneous effects, latent confounders, and history-dependent noise (Peters et al., 2017). The Structural Causal Model (SCM) framework, as implemented in Hyvärinen et al. (2010); Pamfil et al. (2020); Gong et al. (2023); Wang et al. (2024), can theoretically overcome these limitations by explicitly modeling causal relationships between variables. Another line of work (Runge et al., 2019; Runge, 2020a) extends the conditional independence testing-based PC (Spirtes et al., 2000) to time series. However, these methods face scaling and accuracy challenges when applied to

spatiotemporal data due to high dimensionality and spatial correlation effects that can mask true causal relationships between distant locations (Tibau et al., 2022).

**Causal Representation Learning.** The primary objective of causal representation learning is to extract high-level causal variables and their relationships from temporal data. Lippe et al. (2022; 2023) explore this topic specifically in the context of interventional time series data. Meanwhile, Yao et al. (2022b;a); Chen et al. (2024); Morioka & Hyvarinen (2024); Li et al. (2024); Lachapelle et al. (2024) establish identifiability conditions for reconstructing latent time series from observational data. However, their theorems rely on various assumptions about the underlying latent dynamics such as no instantaneous effects, sufficient variability or sparsity. These conditions can be challenging to validate in practice. They also do not consider the spatial structure critical to spatiotemporal causal discovery. Another line of research is Dynamic Causal Modeling (Friston et al., 2003; Stephan et al., 2010; Friston et al., 2013; 2022), which infers causal relationships in dynamical systems and has been applied to neuroscience. However, DCM assumes that the parameters of the forward model (i.e., the relationship between the latent and observed variables) are known *a priori*. In contrast, SPACY guarantees identifiability of latent time series from spatiotemporal data with minimal assumptions about the latent-generating process.

**Spatiotemporal Causal Discovery.** Numerous studies have extended Granger causality to spatiotemporal settings, particularly in climate science (Mosedale et al., 2006; Lozano et al., 2009; Kodra et al., 2011). Another approach to spatiotemporal causal discovery is to perform dimensionality reduction to obtain a smaller number of latent time series and then infer a causal graph among the latent variables (Tibau et al., 2022; Falasca et al., 2024). A key limitation of these methods is that dimensionality reduction occurs independently of the causal structure in the data. Consequently, the latent variables may not correspond to causally relevant entities. Some studies like Sheth et al. (2022) address specific spatial dependencies relevant to the considered domain. Zhao et al. (2023b) extend Yao et al. (2022a) by incorporating spatial structures using graph convolutional networks. Brouillard et al. (2024); Boussard et al. (2023) adopt the single-parent assumption, where each observed variable is influenced by only one latent variable. On the other hand, SPACY allows multiple latent parents per observed variable.

## 3. SPACY: Spatiotemporal Causal Discovery

**Preliminaries.** A Structural Causal Model (Pearl, 2009) (SCM) defines the causal relationships between variables in the form of functional equations. Formally, an SCM over $D$ variables in a time-varying system, with time denoted by $t \in [T]$, consists of a 5-tuple $\langle \mathcal{X}^{(1:T)}, \varepsilon^{(1:T)}, \mathbf{G}, \mathcal{F}, p_{\varepsilon_i^{(1:T)}}(\cdot) \rangle$:

1. Observed variables $\mathcal{X}^{(t)} = \{\mathbf{X}_1^{(t)}, \ldots, \mathbf{X}_D^{(t)}\}$;
2. Noise variables $\varepsilon^{(t)} = \{\varepsilon_1^{(t)}, \ldots, \varepsilon_D^{(t)}\}$ which influence the observed variables;
3. A *Directed Acyclic Graph* (DAG) $\mathbf{G}$, which denotes the causal links among the members of $\mathcal{X}^{(1:T)}$ upto a maximum time-lag of $\tau$.
4. A set of $D$ functions $\mathcal{F} = \{f_1, \ldots, f_D\}$ which determines $\mathcal{X}^{(t)}$ through the equations $\mathbf{X}_i^{(t)} = f_i(\text{Pa}_{\mathbf{G}}^i(\leq t), \varepsilon_i^{(t)})$, where $\text{Pa}_{\mathbf{G}}^i(\leq t) \subset \mathcal{X}^{(t-\tau:t)}$ denotes the parents of node $i$ in $\mathbf{G}$;
5. $p_{\varepsilon_i^{(t)}}$, which describes a distribution over noise $\varepsilon_i^{(t)}$.

**Problem Setting.** We are given $N$ samples of $L$-dimensional time series with $T$ timesteps each. These $L$ time series are arranged in a $K$-dimensional grid of shape $L_1 \times \ldots \times L_K$ such that $L = \prod_{k=1}^K L_k$, with the spatial coordinates scaled to lie in $\mathcal{G} = [0,1]^K$. In our setting, we consider $K = 2$. We denote the observational time series as $\left\{\mathbf{X}_{1:L}^{(1:T),n}\right\}_{n=1}^N$. We assume that the dynamics of the observed data are driven by interactions in a smaller number of *latent* time series. We denote the $D$-dimensional latent time series for each of the $N$ samples as $\left\{\mathbf{Z}_{1:D}^{(1:T),n}\right\}_{n=1}^N$, with $D << L$. The latent time series is stationary with a maximum time lag of $\tau$, meaning the present is influenced by up to $\tau$ past timesteps. Interactions in the latent time series follow an SCM represented by a DAG $\mathbf{G}$. Our goal is to infer the latent time series $\left\{\mathbf{Z}_{1:D}^{(1:T),n}\right\}_{n=1}^N$ and the causal graph $\mathbf{G}$ in an unsupervised manner.

### 3.1. Forward Model

We formalize our assumptions about the data generation process using a probabilistic graphical model (Figure 3). We assume that the latent time series $\mathbf{Z}$ is generated by an SCM with causal graph $\mathbf{G}$. The number of latent variables $D$ is a hyperparameter. The spatial correlations between grid points are captured by the spatial factors

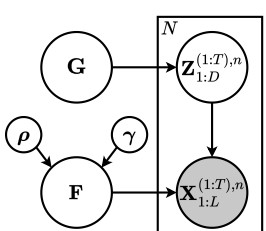

Figure 3: Probabilistic graphical model for SPACY. Shaded circles are observed and hollow circles are latent.

$\mathbf{F} \in \mathbb{R}^{L \times D}$, which map the latent time series $\mathbf{Z}_{1:D}^{(1:T)} \in \mathbb{R}^{D \times T}$ to the observed time series $\mathbf{X}_{1:L}^{(1:T)} \in \mathbb{R}^{L \times T}$. Specifically, the observational time series is generated by applying a grid point-wise non-linearity $g_\ell$, to the product of the spatial factors and latent time series, with additive Gaussian noise. $\ell$ denotes the index of the grid points.

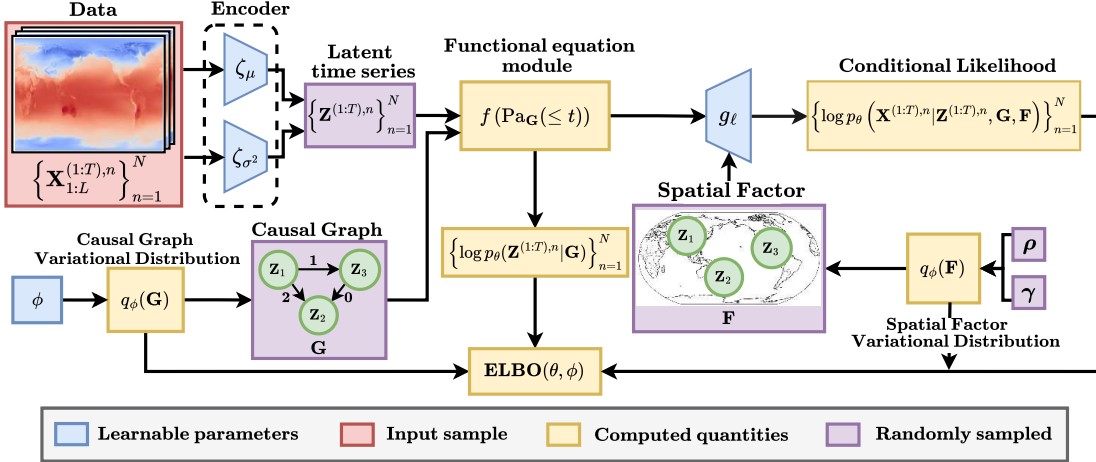

Figure 2: Overview of the ELBO calculation for SPACY. The model processes spatiotemporal data $\left\{\mathbf{X}_{1:L}^{(1:T),n}\right\}_{n=1}^{N}$ to infer latent time series $\left\{\mathbf{Z}_{1:D}^{(1:T),n}\right\}_{n=1}^{N}$, where $D \ll L$. Causal relationships are modeled using a DAG $\mathbf{G}$ sampled from $q_\phi(\mathbf{G})$. Latent time-series are mapped to grid locations via spatial factors $\mathbf{F}$ sampled from $q_\phi(\mathbf{F})$. Arrows in $\mathbf{G}$ are labeled with edge time-lags.

$$\mathbf{X}_\ell^{(t)} = g_\ell\left([\mathbf{FZ}]_\ell^{(t)}\right) + \varepsilon_\ell^{(t)}, \quad \varepsilon_\ell^{(t)} \sim \mathcal{N}(0, \sigma_\ell^2)$$

$$g_\ell(x) = \Xi\left([x, \mathscr{E}_\ell]\right), \quad \mathscr{E}_\ell \in \mathbb{R}^f \tag{1}$$

We implement the nonlinearity $g_\ell$ as an MLP (multilayer perceptron) $\Xi$ shared across all grid-points, with concatenated embeddings $\mathscr{E} \in \mathbb{R}^{L \times f}$ of dimension $f$.

**Latent SCM.** We model the latent SCM describing $\mathbf{Z}^{(t)}$ as an additive noise model (Hoyer et al., 2008):

$$\mathbf{Z}_d^{(t)} = f_d\left(\mathrm{Pa}_\mathbf{G}^d(< t), \mathrm{Pa}_\mathbf{G}^d(t)\right) + \eta_d^{(t)},$$

Note that $\eta_d^{(t)}$ is distinct from the additive Gaussian noise $\varepsilon_\ell^{(t)}$ at the grid-level. The causal graph $\mathbf{G}$ specifies the causal parents of each node, represented by a temporal adjacency matrix with shape $(\tau + 1) \times D \times D$. The parent nodes from previous and current time steps are denoted by $\mathrm{Pa}_\mathbf{G}^d(< t)$ and $\mathrm{Pa}_\mathbf{G}^d(t)$ respectively. We assume that $\mathbf{Z}_d^{(t)}$ is influenced by at most $\tau$ preceding time steps, i.e., $\mathrm{Pa}_\mathbf{G}(< t) \subseteq \{\mathbf{Z}^{(t-1)}, \dots, \mathbf{Z}^{(t-\tau)}\}$. $\mathbf{G}^{(1:\tau)}$ represents the lagged relationships and $\mathbf{G}^{(0)}$ represents the instantaneous edges. The time-lag $\tau$ is treated as a hyperparameter.

In principle, SPACY is compatible with any differentiable temporal causal discovery algorithm. In this work, we choose Rhino (Gong et al., 2023) to model the functional relationships because of its flexibility. It is an identifiable framework that captures both instantaneous effects and history-dependent noise. We parameterize the structural equations $f_d$ using MLPs $\xi_f$ and $\lambda_f$ shared across all nodes. We use trainable embeddings $\mathcal{E} \in \mathbb{R}^{(\tau+1) \times D \times D}$ with em-

bedding dimension $e$ to distinguish between nodes. $f_d$ is defined as:

$$f_d\left(\mathrm{Pa}_\mathbf{G}(\leq t)\right) = \xi_f\left(\sum_{k=0}^\tau \sum_{j=1}^D \mathbf{G}_{j,d}^{(k)} \times \lambda_f\left(\left[\mathbf{Z}_j^{(t-k)}, \mathcal{E}_j^k\right], \mathcal{E}_0^d\right)\right), \tag{2}$$

where $\mathbf{G}_{j,d}^{(k)}$ denotes the presence of an edge from node $j$ to $d$ at the $k^{\text{th}}$ time-lag. The noise model is based on conditional spline flows (Durkan et al., 2019), with the parameters of the spline flow predicted by MLPs $\xi_\eta$ and $\lambda_\eta$, which share a similar architecture to $\xi_f$ and $\lambda_f$.

**Spatial Factors.** The low-dimensional latent time series are mapped to the high-dimensional grid by the spatial factors $\mathbf{F} \in \mathbb{R}^{L \times D}$, where the $d^{\text{th}}$ column represents the influence of the $d^{\text{th}}$ latent variable on each grid location. To effectively capture the correlation between spatially proximate locations under a single latent variable, we model the spatial factors using kernel functions. In theory, any linearly independent, real analytic family of functions would work (see Section 4). In practice, we find that radial basis functions (RBFs) work quite well, following Manning et al. (2014); Sennesh et al. (2020); Farnoosh & Ostadabbas (2021) (see Appendix D.3 for an ablation study). RBFs not only ensure locality, they are also smooth functions that are parameter-efficient. We assume a uniform prior over the grid $\mathcal{G}$ for the center parameter $\rho_d$ of each kernel, and that the scale parameter $\gamma_d$ comes from a standard normal distribution. Mathematically,

$$\rho_d \sim U[0,1]^K, \gamma_d \sim \mathcal{N}(0, I),$$

$$\mathbf{F}_{\ell d} = \mathrm{RBF}_d(x_\ell; \rho_d, \gamma_d) = \exp\left(-\frac{||x_\ell - \rho_d||^2}{\exp(\gamma_d)}\right), \tag{3}$$

where $x_\ell$ denotes the spatial coordinates of the $\ell^{\text{th}}$ grid point.

## 3.2. Variational Inference

Let $\theta$ denote the parameters of the forward model. The likelihood $p_\theta(\mathbf{X})$ is intractable due to the presence of latent variables $\mathbf{Z}$, $\mathbf{G}$ and $\mathbf{F}$. We propose using variational inference, optimizing an evidence lower bound (ELBO) instead.

**Proposition 1.** *The data generation model described in Figure 3 admits the following evidence lower bound (ELBO):*

$$\log p_\theta\left(\mathbf{X}^{(1:T),1:N}\right) \geq \sum_{n=1}^{N} \left\{ \mathbb{E}_{q_\phi(\mathbf{Z}^{(1:T),n}|\mathbf{X}^{(1:T),n})q_\phi(\mathbf{G})q_\phi(\mathbf{F})} \right.$$

$$\left[ \log p_\theta\left(\mathbf{X}^{(1:T),n}|\mathbf{Z}^{(1:T),n}, \mathbf{F}\right) + \left[ \log p_\theta\left(\mathbf{Z}^{(1:T),n}|\mathbf{G}\right) \right.$$

$$\left. \left. - \log q_\phi\left(\mathbf{Z}^{(1:T),n}|\mathbf{X}^{(1:T),n}\right) \right] \right] \right\} - KL\left(q_\phi(\mathbf{G}) \,||\, p(\mathbf{G})\right)$$

$$- KL\left(q_\phi(\mathbf{F}) \,||\, p(\mathbf{F})\right) = ELBO(\theta, \phi) \tag{4}$$

See Section A.1 for the derivation. We outline the computation of the ELBO in Figure 2. $q_\phi$ represents the variational distribution parameterized by $\phi$. The first term $\log p_\theta(\mathbf{X}^{(1:T),n}|\mathbf{Z}^{(1:T),n}, \mathbf{F})$ in equation 4 represents the conditional likelihood of the observed data $\mathbf{X}^{(1:T),n}$ conditioned on $\mathbf{Z}^{(1:T),n}$ and $\mathbf{F}$. The remaining terms represent the KL divergences of the variational distributions from their prior distributions. Next, we describe the design of the variational distributions, with the full implementation details in Appendix B.

**Causal graph** $q_\phi(\mathbf{G})$. The variational distribution of the causal graph is modeled as a product of independent Bernoulli distributions, indicating the presence or absence of an edge.

$$q_\phi(\mathbf{G}) = \prod_{k=0}^{\tau} \prod_{i,j=1}^{D} \text{Bernoulli}\left(\mathcal{W}_{i,j}^k\right),$$

where $\mathcal{W} \in \mathbb{R}^{(\tau+1)\times D \times D}$ is a learned parameter. To estimate the expectation over $q_\phi(\mathbf{G})$, we use Monte Carlo sampling by drawing a single graph sample, employing the Gumbel-Softmax trick (Jang et al., 2017). We ensure that the learned graph $\mathbf{G}$ is a DAG using the acyclicity constraint from Zheng et al. (2018), which is added to the prior $p(\mathbf{G})$.

**Spatial Factor** $q_\phi(\mathbf{F})$. We model the variational distributions of the center $\boldsymbol{\rho}_d$ and scale $\boldsymbol{\gamma}_d$ as normal distributions with learnable mean and log-variance parameters $(\boldsymbol{\mu}_{\boldsymbol{\rho}_d}, v_{\boldsymbol{\rho}_d}), (\boldsymbol{\mu}_{\boldsymbol{\gamma}_d}, v_{\boldsymbol{\gamma}_d})$. To sample from $q_\phi(\mathbf{F})$, we first sample $\boldsymbol{\rho}_d$ and $\boldsymbol{\gamma}_d$ using the reparameterization trick (Kingma & Welling, 2014), and then compute the RBF kernel using these parameters. We ensure that the coordinates of the center lie in $[0,1]^K$ by applying the sigmoid function.

$$\boldsymbol{\rho}_d \sim \mathcal{N}\left(\boldsymbol{\mu}_{\boldsymbol{\rho}_d}, \exp\left(v_{\boldsymbol{\rho}_d}\right) I\right), \boldsymbol{\gamma}_d \sim \mathcal{N}\left(\boldsymbol{\mu}_{\boldsymbol{\gamma}_d}, \exp\left(v_{\boldsymbol{\gamma}_d}\right) I\right)$$

$$\mathbf{F}_{\ell d} = \text{RBF}_d(x_\ell; \boldsymbol{\rho}_d, \boldsymbol{\gamma}_d) = \exp\left(-\frac{||x_\ell - \text{sigmoid}(\boldsymbol{\rho}_d)||^2}{\exp(\boldsymbol{\gamma}_d)}\right)$$

**Latent Time Series** $q_\phi(\mathbf{Z}^{(1:T),n}|\mathbf{X}^{(1:T),n})$. To obtain latents from the observations, we use a neural network encoder. Specifically, we assume $q_\phi(\mathbf{Z}^{(1:T),n}|\mathbf{X}^{(1:T),n})$ to be a normal distribution whose mean and log-variance are parameterized by MLPs $\zeta_\mu$ and $\zeta_{\sigma^2}$.

$$q_\phi\left(\mathbf{Z}^{(1:T),n}|\mathbf{X}^{(1:T),n}\right) = \mathcal{N}\left(\zeta_\mu(\mathbf{X}^{(t),n}), \exp\left(\zeta_{\sigma^2}(\mathbf{X}^{(t),n})\right)\right).$$

We sample the latents $\mathbf{Z}$ from the variational distribution using the reparameterization trick.

## 3.3. Multivariate Extension

In many applications, it is important to model interactions between different variates on the same grid. For instance, understanding how global temperature patterns influence precipitation requires analyzing multivariate relationships. We extend SPACY to handle such multivariate time series data, where the observational time series $\mathbf{X} \in \mathbb{R}^{V \times L \times T}$ consists of $V \geq 1$ variates. We modify SPACY to learn variate-specific spatial factors $\mathbf{F}^{(v)}$ by specifying the number of nodes $D_v$ for each variate $v$. We then combine latent representations from all variates and perform causal discovery with them. For more implementation details, refer to Appendix B.4.

# 4. Identifiability Analysis

We analyze the identifiability of the spatiotemporal generative model from Section 3.1. Identifiability ensures that the latent variables can be uniquely recovered from observations, up to permutation and scaling. Prior work, such as (Yao et al., 2022b;a; Lachapelle et al., 2024), establishes identifiability in temporal settings with nonlinear invertible mixing but uses restrictive assumptions about the latent process, like the absence of instantaneous effects, sparsity of the causal graph, or sufficient variability. In contrast, we demonstrate identifiability in our spatiotemporal setting, where the observations are a nonlinear function of the product of latent time series and spatial factors, without relying on such assumptions, by explicitly leveraging spatial correlations. This is made possible by the overdetermined nature of the problem, where the number of spatial locations exceeds the latent dimension, allowing for identifiability with minimal assumptions about the latent process.

Specifically, we generalize the discrete grid $\mathcal{G}$ to a continuous spatial domain with infinite spatial resolution. This generalization allows us to the prove identifiability condition using tools from real analysis. Specifically, for every point $\ell$ in the $K$-dimensional spatial domain $\mathcal{G} = (0,1)^K$, we observe a corresponding time series $\mathbf{X}(\ell)$, representing a $T$-dimensional random variable. We model the spatial

factors as function evaluations of a family of linearly independent functions. Notably, the family of RBF functions is one such family of functions (Smola & Schölkopf, 1998). We introduce the notion of a spatial factor process (SFP), and mathematically describe the identifiability of SFPs.

**Definition 1** (Spatial Factor Process). Let $\mathcal{G} = (0,1)^K$ be a $K-$dimensional spatial domain, and let $\{\mathbf{Z}^{(t)}\}_{t=1}^T$ denote the latent causal process, where $\mathbf{Z}^{(t)} \in \mathbb{R}^D$. Let $\mathcal{F} = \{F_{\psi_1}, ..., F_{\psi_D}\}$ be a finite linearly independent family of functions defined on $\mathcal{G}$, and $\mathscr{G} = \{g_\ell \colon \mathbb{R} \to \mathbb{R} \mid \ell \in \mathcal{G}\}$ be a family of functions defined for each point $\ell$ on the grid $\mathcal{G}$. Let $p_{\varepsilon_\ell^{(t)}}$ be a zero-mean noise distribution. The Spatial Factor Process SFP $\left(\left\{\mathbf{Z}^{(t)}\right\}_{t=1}^T, \mathcal{F}, \mathscr{G}, p_{\varepsilon_\ell^{(t)}}\right)$, denoted by $\mathbf{X}$, is defined as follows: For each location $\ell \in \mathcal{G}$ on the grid and $t \in [T]$,

$$\mathbf{X}^{(t)}(\ell) = g_\ell\left(\mathbf{F}_\ell^\top \mathbf{Z}^{(t)}\right) + \varepsilon_\ell^{(t)}, \qquad (5)$$

where

$$\mathbf{F}_\ell = \left[F_{\psi_1}(\ell), \dots, F_{\psi_D}(\ell)\right]^\top.$$

We can define the identifiability of such SFPs as below:

**Definition 2** (Identifiability of SFPs). Let $\mathbf{X}$ denote the true generative SFP, specified by $\left(\left\{\mathbf{Z}^{(t)}\right\}_{t=1}^T, \{F_{\psi_i}\}_{i=1}^D, \{g_\ell \mid \ell \in \mathcal{G}\}, p_{\varepsilon_\ell^{(t)}}\right)$ with observational distribution $p(\mathbf{X}^{(t)}|\mathbf{Z}^{(t)}; \mathbf{F})$, where

$$\mathbf{X}^{(t)}(\ell) = g_\ell\left(\mathbf{F}_\ell^\top \mathbf{Z}^{(t)}\right) + \varepsilon_\ell^{(t)}, \qquad (6)$$

with $\varepsilon_\ell^{(t)} \sim p_{\varepsilon_\ell^{(t)}}$ for some zero-mean noise distribution $p_{\varepsilon_\ell^{(t)}}$. Suppose we have a learned SFP $\widehat{\mathbf{X}}$, specified by $\left(\left\{\widehat{\mathbf{Z}}^{(t)}\right\}_{t=1}^T, \left\{F_{\widehat{\psi}_i}\right\}_{i=1}^D, \{\widehat{g}_\ell \mid \ell \in \mathcal{G}\}, p_{\varepsilon_\ell^{(t)}}\right)$ with observational distribution $p(\widehat{\mathbf{X}}^{(t)}|\widehat{\mathbf{Z}}^{(t)}; \widehat{\mathbf{F}})$, where

$$\widehat{\mathbf{X}}^{(t)}(\ell) = \widehat{g}_\ell\left(\widehat{\mathbf{F}}_\ell^\top \widehat{\mathbf{Z}}^{(t)}\right) + \widehat{\varepsilon}_\ell^{(t)}, \qquad (7)$$

with $\widehat{\varepsilon}_\ell^{(t)} \sim p_{\varepsilon_\ell^{(t)}}$. The latent process $\{\mathbf{Z}^{(t)}\}$ is said to be *identifiable* upto permutation and scaling, if:

$$p\left(\mathbf{X}^{(t)}(\ell)|\mathbf{Z}^{(t)}; \mathbf{F}_\ell\right) = p\left(\widehat{\mathbf{X}}^{(t)}(\ell)|\widehat{\mathbf{Z}}^{(t)}; \widehat{\mathbf{F}}_\ell\right), \forall \ell \in \mathcal{G}, t \in [T]$$

$$\implies \widehat{\mathbf{Z}}_t = PS\mathbf{Z}^{(t)}, \text{ and } \{F_{\psi_i}\}_{i=1}^D = \left\{F_{\widehat{\psi}_i}\right\}_{i=1}^D,$$

for some permutation matrix $P$ and scaling matrix $S$.

We first prove the identifiability of SFPs in the absence of nonlinearity, ($g_\ell = \text{Id}$) by leveraging the linear independence of the spatial factors $\mathcal{F}$. We then extend the proof to the general setting with nonlinear, invertible mapping, with real analytic spatial factor functions.

**Theorem 1** (Identifiability of Linear SFPs). *Suppose we are given two SFPs* $\mathbf{X} = SFP\left(\mathbf{Z}, \mathcal{F}, \{g_\ell \mid \ell \in \mathcal{G}\}, p_{\varepsilon_\ell^{(t)}}\right)$ *and* $\widehat{\mathbf{X}} = SFP\left(\widehat{\mathbf{Z}}, \widehat{\mathcal{F}}, \{\widehat{g}_\ell \mid \ell \in \mathcal{G}\}, p_{\varepsilon_\ell^{(t)}}\right)$ *specified by Equations 6 and 7 respectively, such that both $\mathcal{F}$ and $\widehat{\mathcal{F}}$ belong to the same (potentially infinite) family of linearly independent functions. Further, suppose that the following conditions are satisfied:*

1. ***Linearity:*** *For all $\ell \in \mathcal{G}$, $g_\ell = \widehat{g}_\ell = Id$, where Id is the identity function.*
2. ***Non-degenerate latent processes:*** *For all $d \in [D]$, $\exists t_0 \in [T]$ such that $\mathbf{Z}_d^{(t_0)} \neq 0$, that is, none of the time series is trivially zero. A similar condition holds for $\widehat{\mathbf{Z}}$.*
3. ***Characteristic function of the noise:*** *For all $\ell \in \mathcal{G}$ and $t \in [T]$, the set $\left\{x \in \mathbb{R} \mid \varphi_{\varepsilon_\ell^{(t)}}(x) = 0\right\}$ has measure zero where $\varphi_{\varepsilon_\ell^{(t)}}$ represents the characteristic function of the density $p_{\varepsilon_\ell^{(t)}}$.*

*If $p(\mathbf{X}^{(t)}(\ell)|\mathbf{Z}^{(t)}; \mathbf{F}_\ell) = p\left(\widehat{\mathbf{X}}^{(t)}(\ell)|\widehat{\mathbf{Z}}^{(t)}; \widehat{\mathbf{F}}_\ell\right)$ for every $\ell \in \mathcal{G}$ and $t \in [T]$, then $\mathbf{Z} = P\tilde{\mathbf{Z}}$ and $\mathcal{F} = \widehat{\mathcal{F}}$ for some permutation matrix $P$.*

We now consider the general case where the functions $g_\ell$ may be nonlinear. When the spatial factors are analytic—meaning they can be represented by power series everywhere— we can construct invertible maps between $\mathbf{Z}^{(t)}$ and $\widehat{\mathbf{Z}}^{(t)}$. Notably, these maps can be shown to have Jacobian matrices that are permutation and scaling matrices.

**Theorem 2** (Identifiability of General SFPs). *Suppose we are given two SFPs* $\mathbf{X} = SFP\left(\mathbf{Z}, \mathcal{F}, \{g_\ell \mid \ell \in \mathcal{G}\}, p_{\varepsilon_\ell^{(t)}}\right)$ *and* $\widehat{\mathbf{X}} = SFP\left(\widehat{\mathbf{Z}}, \widehat{\mathcal{F}}, \{\widehat{g}_\ell \mid \ell \in \mathcal{G}\}, p_{\varepsilon_\ell^{(t)}}\right)$ *specified by Equations 6 and 7 respectively, such that both $\mathcal{F}$ and $\widehat{\mathcal{F}}$ belong to the same (potentially infinite) family of linearly independent functions. Further, suppose that the following conditions are satisfied:*

1. ***Diffeomorphisms:*** *For all $\ell \in \mathcal{G}$, $g_\ell, \widehat{g}_\ell$ are diffeomorphisms, that is, $g_\ell, \widehat{g}_\ell$ are invertible and continuously differentiable, and their inverses are also continuously differentiable.*
2. ***Real analytic functions:*** *All the functions $F_{\psi_i} \in \mathcal{F}, F_{\widehat{\psi}_i} \in \widehat{\mathcal{F}}$ are real analytic.*
3. ***Non-degenerate latent processes:*** *For all $d \in [D]$, $\exists t_0 \in [T]$ such that $\mathbf{Z}_d^{(t_0)} \neq 0$, that is, none of the time series is trivially zero. A similar condition holds for $\widehat{\mathbf{Z}}$.*
4. ***Characteristic function of the noise:*** *For all $\ell \in \mathcal{G}$ and $t \in [T]$, the set $\left\{x \in \mathbb{R} \mid \varphi_{\varepsilon_\ell^{(t)}}(x) = 0\right\}$ has measure zero where $\varphi_{\varepsilon_\ell^{(t)}}$ represents the characteristic function of the density $p_{\varepsilon_\ell^{(t)}}$.*

If $p(\mathbf{X}^{(t)}(\ell)|\mathbf{Z}^{(t)};\mathbf{F}_\ell) = p\left(\widehat{\mathbf{X}}^{(t)}(\ell)|\widehat{\mathbf{Z}}^{(t)};\widehat{\mathbf{F}}_\ell\right)$ *for every* $\ell \in \mathcal{G}$ *and* $t \in [T]$, *then* $\mathbf{Z} = PS\tilde{\mathbf{Z}}$ *and* $\mathcal{F} = \widehat{\mathcal{F}}$ *for some permutation matrix* $P$ *and scaling matrix* $S$.

The detailed mathematical statements and proofs for these results are provided in Appendix A.2.

**Applicability to finite grids.** While our identifiability theory assumes a continuous spatial domain, the key insights apply to finite grids if the discretization is dense, that is, the number of grid points is sufficiently high. If the number of grid points $L >> D$, the system is overdetermined, and the redundancy can be used to uniquely identify the latent factors. Measure-zero pathologies which prevent identifiability in the continuous case, remain probabilistically unlikely in finite grids. Our experiments confirm these insights, showing accurate recovery of latent factors in practice.

**Recovery of the causal graph.** Theorems 1 and 2 establish the identifiability of the latent variables from observational data, up to permutation and scaling. Once these latent processes are recovered, we can apply causal discovery algorithms with identifiability guarantees—such as Rhino, which we use in this work—to infer the causal graph among the latent variables, provided that the algorithm's identifiability conditions are met. This is possible because the causal relationships are encoded in the conditional independence relationships of the latent time series. For completeness, we list the assumptions of Rhino in Appendix A.3.

# 5. Experiments

We assess SPACY's ability to capture causal relationships across various spatiotemporal settings using both synthetic datasets with known ground truth and simulated climate datasets. Our results demonstrate that SPACY consistently uncovers accurate causal relationships while generating interpretable outputs.

**Baselines.** We compare SPACY with state-of-the-art baselines. We include the two-step algorithms Mapped PCMCI (Varimax-PCA + PCMCI$^+$ with Partial Correlation test) (Tibau et al., 2022; Runge, 2020b) and the Linear Response method (Falasca et al., 2024). We also evaluate against the causal representation learning approaches, LEAP (Yao et al., 2022b), TDRL (Yao et al., 2022a), and CDSD (Boussard et al., 2023; Brouillard et al., 2024)

## 5.1. Synthetic Data

**Setup.** Since real-world datasets lack ground truth causal graphs, we generate synthetic datasets with known causal relationships to benchmark SPACY's causal discovery performance. These are generated from randomly constructed ground-truth graphs and follow the forward model described

in Figure 3. We experiment with several configurations of synthetic data. The latent time series are generated using either (1) a linear structural causal model (SCM) with randomly initialized weights and additive Gaussian noise, or (2) a nonlinear SCM, where the structural equations are modeled by randomly initialized MLPs, combined with additive history-dependent conditional-spline noise. The mapping function $g_\ell$ is set as (1) linear, where the identity function is used, or (2) nonlinear, where an MLP is used. The spatial factors are constructed using RBF kernels with randomly initialized center and scale parameters, and the Euclidean distance is used as the underlying metric. For each configuration, we generate $N = 100$ samples, each with time length $T = 100$ and a grid of size $100 \times 100$ ($L = 10^4$). Datasets are generated with $D = 10, 20$ and $30$ nodes in each setting. For more details on dataset generation, refer to Appendix C.1.1.

We assess the performance of SPACY and the baselines using two metrics: the orientation F1 score of the inferred causal graph, and the mean correlation coefficient (MCC) between the learned and ground-truth latents. More details on evaluation are presented in Appendix B.6.

**Results.** The results of the synthetic experiments are shown in Figure 4. SPACY consistently outperforms all other methods across all settings of $D$ in terms of F1 score. On the linear SCM datasets, SPACY remains slightly ahead of CDSD and Mapped PCMCI, while LEAP, TDRL, and Linear-Response exhibit weaker performance. However, SPACY significantly outperforms the baselines in the nonlinear settings. A pronounced performance drop is observed for LEAP, TDRL, and Linear-Response, whose F1 scores decline sharply as $D$ increases. SPACY's performance scales more effectively with increasing $D$, further widening the gap in performance.

The recovery of the latent variables, measured by the MCC score, follows a similar pattern. SPACY achieves similar or better MCC scores compared to CDSD and Mapped PCMCI, while LEAP, TDRL, and Linear-Response consistently show lower MCC scores across all configurations. Figure 8 provides a visual illustration of the recovered spatial factors.

**Scalability.** We also measure the scalability of SPACY with increasing grid-size. For this experiment, we used the dataset with linear SCM and nonlinear spatial mapping $g_\ell$, and varied the grid size from $L = 30 \times 30$ to $L = 250 \times 250$. Figure 5 demonstrates the scalability and performance of SPACY compared to the baseline methods as the grid size $L$ increases. The runtime plot indicates that, while all methods experience an increase in runtime with increasing grid size, SPACY strikes a good balance, exhibiting moderate growth in computational time while maintaining strong causal discovery performance. Although Mapped-PCMCI is the most efficient in terms of runtime, it underperforms in causal

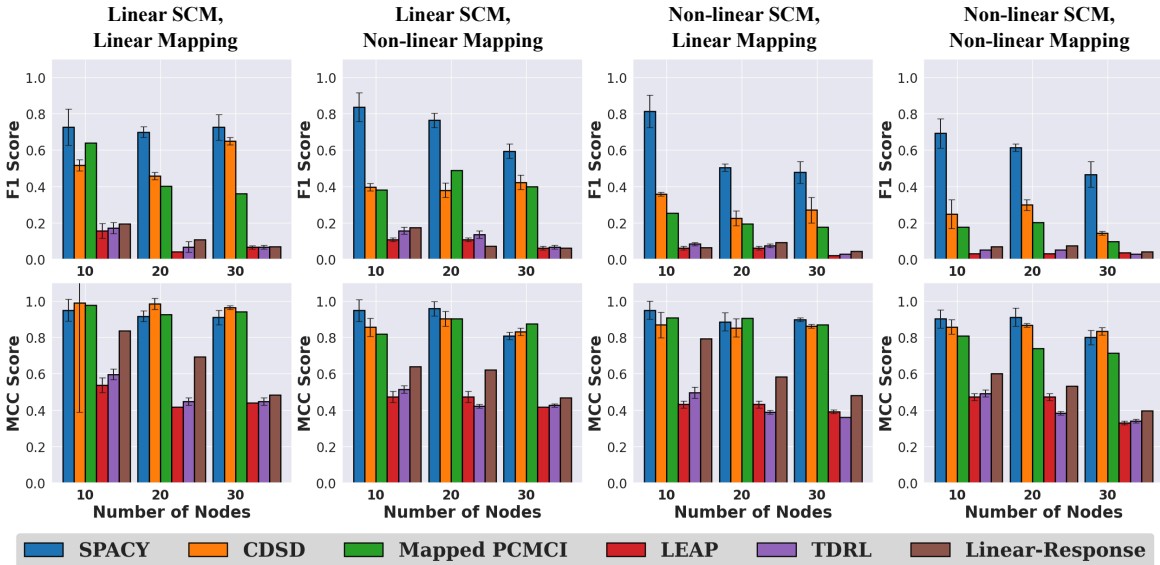

Figure 4: Results on different configurations of the synthetic datasets. We report the F1 and MCC scores for each method across different latent dimensions $D$. Average over 5 runs reported

discovery. LEAP, TDRL, and CDSD show similar or higher computational costs than SPACY but fail to match its performance. Linear-Response, in particular, scales poorly in terms of runtime with increasing grid size. Additional results and ablation studies are in Appendix D.

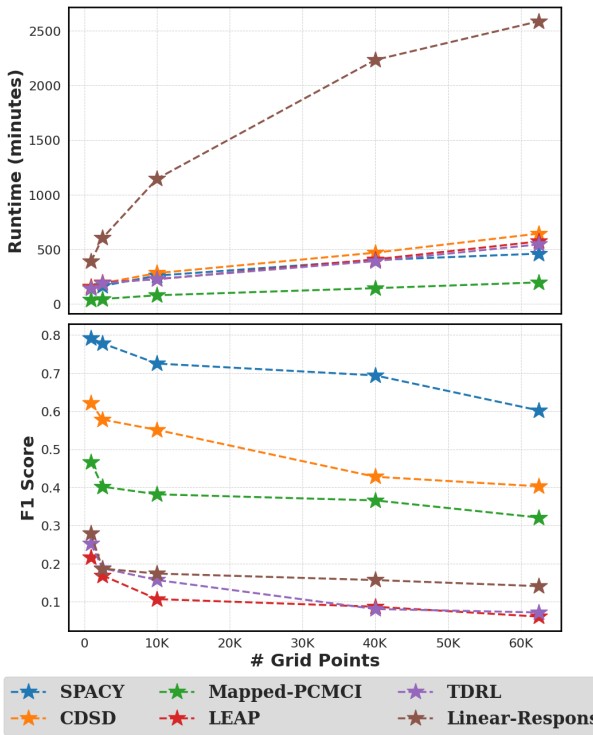

Figure 5: Runtime (in minutes) (top) and F1 score (bottom) across different grid sizes. Average over 5 runs reported.

## 5.2. Climate Dataset

The Global Climate Dataset (Baker et al., 2019) is a mixed real-simulated dataset containly monthly global temperature and precipitation from 1999 to 2001. For more details about the dataset and preprocessing steps, refer to Appendix D.4. In our experiments, we use the RBF kernel with the Haversine distance as the metric, as detailed in Appendix B.3.1

**Results.** In the absence of a ground truth causal graph, we qualitatively evaluate the spatial factors and causal graph inferred by SPACY. We highlight several spatial modes identified by SPACY that correspond to critical regions that significantly influence global climate patterns, including coastlines of major land masses (e.g., East Asia, Northern Europe) and key ocean areas (e.g., Central Pacific, South Atlantic). We refer the interested reader to Figure 15 in Appendix C.4 for the full visualization of the spatial factors and causal graphs inferred by SPACY from this dataset.

Figure 6 highlights three subgraphs of the inferred graph, corresponding to the Madden-Julian Oscillation (MJO) (Madden & Julian, 1971; 1972), Northern Atlantic Oscillation (NAO) (Hurrell et al., 2003; Hurrell, 1995; Chen & den Dool, 2003), and Antarctic Oscillation (AAO) (Thompson & Solomon, 2002; Mo, 2000). SPACY discovers causal links consistent with known teleconnection mechanisms: short-lagged connections (1–2 months) between mid-Pacific (nodes 0 and 17) and Southeast Asian precipitation/temperature nodes (nodes 2 and 11) align with the MJO's eastward-propagating convection and 30–60 day cycle. Similarly, simultaneous precipitation links between Northeast America (nodes 8) and Western Europe (node 1), alongside dipole-like connections in Europe/North Africa

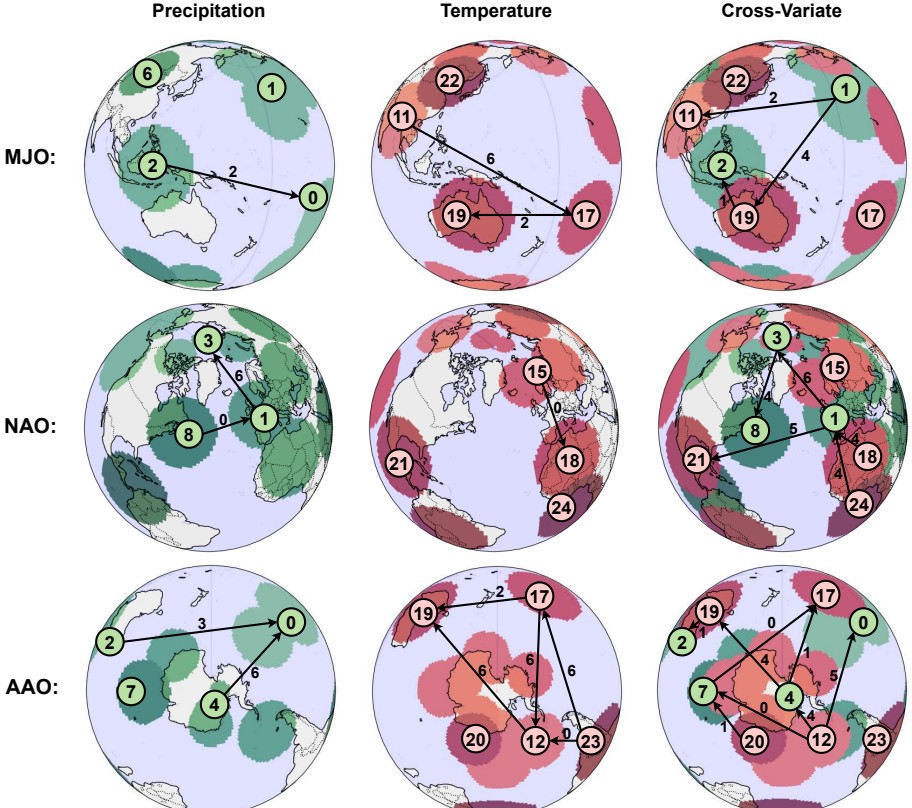

Figure 6: **Visualization of discovered causal relationships in climate datasets.** Subgraph among regions associated with the (top) Madden-Julian Oscillation, (middle) Northern Atlantic Oscillation, (bottom) Antarctic Oscillation. Causal links are shown for (left) Precipitation, (middle) Temperature, and (right) cross-variate interactions. Numbers on the edges indicate time-lag, while the numbers on the nodes indicate the node indices.

(nodes 15 and 18), mirror the NAO's temperature and precipitation dipole. The AAO subgraph captures Antarctic-centered teleconnections, including causal ties among Australia (nodes 2 and 19), Antarctica (nodes 4, 7 and 12), and South America (node 23), reflecting the AAO's hemispheric influence on Southern Ocean dynamics. In all these cases, SPACY infers nodes that are spatially confined with clear boundaries. In contrast, Mapped PCMCI (Figure 16) produces broadly distributed components that are challenging to interpret.

It is important to note that while the isotropic RBF kernel represents a modeling simplification, key atmospheric processes—such as the Walker circulation, monsoonal systems, and ENSO teleconnections—are inherently localized due to heterogeneous boundary conditions (e.g., land-sea contrast, topography) and regional forcings. Crucially, SPACY successfully resolves these localized relationships despite its kernel assumptions, achieving greater physical interpretability than traditional methods. Approaches relying on principal components, for instance, often obscure spatial coherence through domain-wide averaging, whereas SPACY preserves geographically anchored causal links tied to identifiable mechanisms. This localization advantage allows

the framework to avoid diffuse, unphysical spatial factors common in conventional teleconnection analyses.

## 6. Conclusion

In this work, we tackled the challenge of inferring causal relationships from high-dimensional spatiotemporal data. We introduced SPACY, an end-to-end framework based on variational inference, designed to learn latent causal representations and the underlying structural causal model. SPACY addressed the issue of high dimensionality by identifying causal structures in a low-dimensional latent space. It aggregated spatially correlated grid points through kernels, mapping observed time series into these latent representations. We also established a novel identifiability result for the model at infinite spatial resolution, leveraging tools from real analysis. On synthetic datasets, SPACY outperformed baselines and successfully recovered causal links in challenging settings where existing approaches failed. Applied to real-world climate data, SPACY uncovered established patterns from climate science, identifying teleconnections linked to phenomena such as the Madden-Julian Oscillation, the North Atlantic Oscillation, and the Antarctic Oscillation.

## Acknowledgement

This work was supported in part by the U.S. Army Research Office under Army-ECASE award W911NF-07-R-0003-03, the U.S. Department Of Energy, Office of Science, IARPA HAYSTAC Program, NSF Grants SCALE MoDL-2134209, CCF-2112665 (TILOS), #2205093, #2146343, and #2134274, as well as CDC-RFA-FT-23-0069 from the CDC's Center for Forecasting and Outbreak Analytics. Kun Wang acknowledges support from the HDSI Undergraduate Research Scholarship Program.

## Impact Statement

This paper presents work whose goal is to advance the field of Machine Learning. There are many potential societal consequences of our work, none which we feel must be specifically highlighted here.

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

# Appendix to

# "Discovering Latent Causal Graph from Spatiotemporal Data"

## A. Theory

### A.1. ELBO Derivation

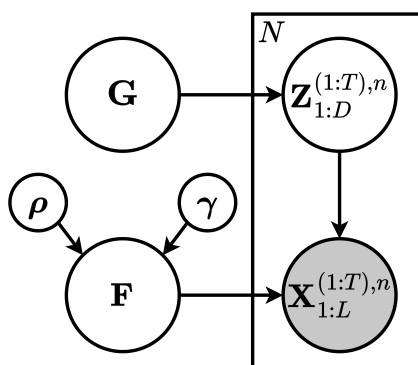

$$\mathbf{Z}_d^{(t)} = f_d\left(\mathrm{Pa}_G^d(<t), \mathrm{Pa}_G^d(t)\right) + \eta_d^{(t)}$$
$$\boldsymbol{\rho}_d \sim U[0,1]^K, \boldsymbol{\gamma}_d \sim \mathcal{N}(0, I)$$
$$\mathbf{F}_d = [\mathrm{RBF}_d(x_\ell; \boldsymbol{\rho}_d, \boldsymbol{\gamma}_d)]_{\ell=1}^L, \quad x_\ell \in \mathcal{G}$$
$$\mathbf{X}_\ell = g_\ell\left([\mathbf{FZ}]_\ell\right) + \varepsilon_\ell$$
$$\varepsilon_\ell \sim \mathcal{N}(0, \sigma_\ell^2 I)$$

Figure 7: Probabilistic graphical model for SPACY and the generative equations. Shaded circles are observed and hollow circles are latent.

**Proposition 1.** *The data generation model described in Figure 3 admits the following evidence lower bound (ELBO):*

$$\log p_\theta\left(\mathbf{X}^{(1:T),1:N}\right) \geq \sum_{n=1}^N \left\{ \mathbb{E}_{q_\phi(\mathbf{Z}^{(1:T),n}|\mathbf{X}^{(1:T),n})q_\phi(\mathbf{G})q_\phi(\mathbf{F})}\left[\log p_\theta\left(\mathbf{X}^{(1:T),n}|\mathbf{Z}^{(1:T),n}, \mathbf{F}\right)\right.\right.$$
$$\left.\left. + \left[\log p_\theta\left(\mathbf{Z}^{(1:T),n}|\mathbf{G}\right) - \log q_\phi(\mathbf{Z}^{(1:T),n}|\mathbf{X}^{(1:T),n})\right]\right]\right\} + \mathbb{E}_{q_\phi(\mathbf{G})}[\log p(\mathbf{G}) - \log q_\phi(\mathbf{G})]$$
$$+ \mathbb{E}_{q_\phi(\mathbf{F})}[\log p(\mathbf{F}) - \log q_\phi(\mathbf{F})] = ELBO(\theta, \phi)$$

*Proof.* We begin with the log-likelihood of the observed data:

$$\log p_\theta\left(\mathbf{X}^{(1:T),1:N}\right) = \log \int p_\theta\left(\mathbf{X}^{(1:T),1:N}, \mathbf{Z}^{(1:T),1:N}, \mathbf{G}, \mathbf{F}\right) d\mathbf{Z}\, d\mathbf{G}\, d\mathbf{F}$$

We multiply and divide by the variational distribution $q_\phi\left(\mathbf{Z}^{(1:T),1:N}|\mathbf{X}^{(1:T),1:N}\right) q_\phi(\mathbf{G}) q_\phi(\mathbf{F})$ to create an evidence lower bound (ELBO) using Jensen's inequality:

$$\log p_\theta\left(\mathbf{X}^{(1:T),1:N}\right)$$
$$= \log \int \frac{q_\phi\left(\mathbf{Z}^{(1:T),1:N}|\mathbf{X}^{(1:T),1:N}\right) q_\phi(\mathbf{G}) q_\phi(\mathbf{F})}{q_\phi\left(\mathbf{Z}^{(1:T),1:N}|\mathbf{X}^{(1:T),1:N}\right) q_\phi(\mathbf{G}) q_\phi(\mathbf{F})} p_\theta\left(\mathbf{X}^{(1:T),1:N}, \mathbf{Z}^{(1:T),1:N}, \mathbf{G}, \mathbf{F}\right) d\mathbf{Z}\, d\mathbf{G}\, d\mathbf{F}$$
$$\geq \mathbb{E}_{q_\phi(\mathbf{Z}^{(1:T),1:N}|\mathbf{X}^{(1:T),1:N})q_\phi(\mathbf{G})q_\phi(\mathbf{F})}\left[\log \frac{p_\theta\left(\mathbf{X}^{(1:T),1:N}, \mathbf{Z}^{(1:T),1:N}, \mathbf{G}, \mathbf{F}\right)}{q_\phi\left(\mathbf{Z}^{(1:T),1:N}|\mathbf{X}^{(1:T),1:N}\right) q_\phi(\mathbf{G}) q_\phi(\mathbf{F})}\right]. \tag{8}$$

By the assumptions of the data generative process,

$$p_\theta\left(\mathbf{X}^{(1:T),1:N}, \mathbf{Z}^{(1:T),1:N}, \mathbf{G}, \mathbf{F}\right) = p_\theta\left(\mathbf{X}^{(1:T),1:N}|\mathbf{Z}^{(1:T),1:N}, \mathbf{F}\right) p_\theta\left(\mathbf{Z}^{(1:T),1:N}|\mathbf{G}\right) p(\mathbf{F}) p(\mathbf{G})$$

Further, note that $\mathbf{X}^{(1:T),1:N}$ are conditionally independent given $\mathbf{F}, \mathbf{Z}^{(1:T),1:N}$. Also, $\mathbf{X}^{(1:T),n}$ is conditionally independent of $\mathbf{Z}^{(1:T),m}$ given $\mathbf{Z}^{(1:T),n}, \mathbf{F}$ for $m \neq n$. This implies that:

$$p_\theta\left(\mathbf{X}^{(1:T),1:N}|\mathbf{Z}^{(1:T),1:N}, \mathbf{F}\right) = \prod_{n=1}^N p_\theta\left(\mathbf{X}^{(1:T),n}|\mathbf{Z}^{(1:T),n}, \mathbf{F}\right).$$

Similarly, $\mathbf{Z}^{(1:T),1:N}$ are conditionally independent given $\mathbf{G}$, which implies

$$p_\theta\left(\mathbf{Z}^{(1:T),1:N}|\mathbf{G}\right) = \prod_{n=1}^N p_\theta\left(\mathbf{Z}^{(1:T),n}|\mathbf{G}\right).$$

Substituting these terms back into equation 8 and grouping terms according to the variables $\mathbf{Z}, \mathbf{G}, \mathbf{F}$ yields the ELBO.

$$\log p_\theta\left(\mathbf{X}^{(1:T),1:N}\right) \geq \sum_{n=1}^N \left\{ \mathbb{E}_{q_\phi(\mathbf{Z}^{(1:T),n}|\mathbf{X}^{(1:T),n})q_\phi(\mathbf{G})q_\phi(\mathbf{F})}\left[\log p_\theta\left(\mathbf{X}^{(1:T),n}|\mathbf{Z}^{(1:T),n}, \mathbf{G}, \mathbf{F}\right)\right.\right.$$
$$\left.\left. + \left(\log p_\theta\left(\mathbf{Z}^{(1:T),n}|\mathbf{G}\right) - \log q_\phi\left(\mathbf{Z}^{(1:T),n}|\mathbf{X}^{(1:T),n}\right)\right)\right]\right\}$$
$$+ \mathbb{E}_{q_\phi(\mathbf{G})}\left[\log p(\mathbf{G}) - \log q_\phi(\mathbf{G})\right]$$
$$+ \mathbb{E}_{q_\phi(\mathbf{F})}\left[\log p(\mathbf{F}) - \log q_\phi(\mathbf{F})\right] \equiv \text{ELBO}(\theta, \phi).$$

$\square$

## A.2. Identifiability

In this work, we extend the notion of identifiability in latent variable models to the spatiotemporal setting considered in this paper. Informally, a latent variable model is said to be identifiable if the underlying latent variables can be uniquely recovered from observations up to permissible ambiguities such as permutation or scaling.

We begin by formalizing the notion of a spatiotemporal process over a continuous spatial domain, which can be thought about as a gridded time series in a grid with infinite resolution. This abstraction enables us to reason about these systems using tools from real analysis. Although our theory assumes a continuous domain, the insights extend to discrete grids with dense discretization, that is, $L >> D$.

**Definition 0** (Linearly Independent Family of Functions). Let $\mathcal{F}$ be a family of real-valued, parametric functions $\mathcal{F} = \left\{f_\psi \mid \mathbb{R}^K \to \mathbb{R}\right\}$. $\mathcal{F}$ is said to be a linearly independent family if, for any finite set $\{\psi_1, ..., \psi_n\}$, we have

$$\sum_{k=1}^n \alpha_k f_{\psi_k} = 0 \implies \alpha_k = 0 \quad \forall k \in [n]. \tag{9}$$

**Definition 1** (Spatial Factor Process). Let $\mathcal{G} = (0,1)^K$ be a $K-$dimensional spatial domain, and $\mathcal{F} = \{F_{\psi_1}, ..., F_{\psi_D}\}$ be a finite linearly independent family of functions defined on $\mathcal{G}$. Denote $\mathscr{G} = \left\{g_\ell\colon \mathbb{R} \to \mathbb{R} \mid \ell \in \mathcal{G}\right\}$ as a family of functions defined for each point $\ell$ on the grid $\mathcal{G}$. Let $\{\mathbf{Z}^{(t)}\}_{t=1}^T, \mathbf{Z}^{(t)} \in \mathbb{R}^D$ denote the latent causal process and $p_{\varepsilon_\ell^{(t)}}$ be a zero-mean noise distribution. For each location $\ell \in \mathcal{G}$ and time $t \in [T]$, we assume the observation $\mathbf{X}$ follows the *Spatial Factor Process* $\text{SFP}\left(\left\{\mathbf{Z}^{(t)}\right\}_{t=1}^T, \mathcal{F}, \mathscr{G}, p_{\varepsilon_\ell^{(t)}}\right)$ defined below:

$$\mathbf{X}^{(t)}(\ell) = g_\ell\left(\mathbf{F}_\ell^\top \mathbf{Z}^{(t)}\right) + \varepsilon_\ell^{(t)}, \tag{10}$$

where

$$\mathbf{F}_\ell = \begin{bmatrix} F_{\psi_1}(\ell) \\ \vdots \\ F_{\psi_D}(\ell) \end{bmatrix}.$$

A *Spatial Factor Process* (SFP) generalizes gridded time series to a continuous spatial domain $\mathcal{G} = (0,1)^K$, where observations are modeled at *all* points in the grid. The dynamics are driven by a latent process $\{\mathbf{Z}^{(t)}\}$, while spatial structure is captured through factors $\mathbf{F}_\ell = [F_{\psi_1}(\ell), \ldots, F_{\psi_D}(\ell)]^\top$, composed of linearly independent functions evaluated at each $\ell \in \mathcal{G}$. Location-specific nonlinearities are modeled using mappings $g_\ell(\cdot)$, which transform the latent factors $\mathbf{Z}^{(t)}$ into observations at each point.

The identifiability question of SFP is: given the observational distribution $p\left(\mathbf{X}^{(t)}|\mathbf{Z}^{(t)}; \mathbf{F}\right)$ generated from a latent causal process $\{\mathbf{Z}^{(t)}\}_{t=1}^T$, can we uniquely recover the latents (Yao et al., 2022b;a; Moran et al., 2022)? In other words, if we learn a generative model $p\left(\widehat{\mathbf{X}}^{(t)}|\widehat{\mathbf{Z}}^{(t)}; \widehat{\mathbf{F}}\right)$ such that $p(\mathbf{X}|\mathbf{Z}^{(t)}; \mathbf{F}) = p(\widehat{\mathbf{X}}^{(t)}|\widehat{\mathbf{Z}}^{(t)}; \widehat{\mathbf{F}})$, can we infer that $\mathbf{Z}^{(t)} = \widehat{\mathbf{Z}}^{(t)}$? In practice,

we seek recovery of the latents up to trivial transformations like permutation or scaling. We formalize this notion with the following definition.

**Definition 2** (Identifiability of SFPs)**.** Let $\mathbf{X}$ denote the true generative SFP, specified by $\left( \left\{ \mathbf{Z}^{(t)} \right\}_{t=1}^{T}, \{F_{\psi_i}\}_{i=1}^{D}, \{g_\ell \mid \ell \in \mathcal{G}\}, p_{\varepsilon_\ell^{(t)}} \right)$ (as described in Definition 1) with observational distribution $p(\mathbf{X}^{(t)}|\mathbf{Z}^{(t)}; \mathbf{F})$, where

$$\mathbf{X}^{(t)}(\ell) = g_\ell \left( \mathbf{F}_\ell^\top \mathbf{Z}^{(t)} \right) + \varepsilon_\ell^{(t)}, \tag{11}$$

with $\varepsilon_\ell^{(t)} \sim p_{\varepsilon_\ell^{(t)}}$ for some zero-mean noise distribution $p_{\varepsilon_\ell}$. Suppose we have a learned SFP $\widehat{\mathbf{X}}$, specified by $\left( \left\{ \widehat{\mathbf{Z}}^{(t)} \right\}_{t=1}^{T}, \left\{ F_{\widehat{\psi_i}} \right\}_{i=1}^{D}, \{\widehat{g}_\ell \mid \ell \in \mathcal{G}\}, p_{\varepsilon_\ell^{(t)}} \right)$ with observational distribution $p(\widehat{\mathbf{X}}^{(t)}|\widehat{\mathbf{Z}}^{(t)}; \widehat{\mathbf{F}})$, where

$$\widehat{\mathbf{X}}^{(t)}(\ell) = \widehat{g}_\ell \left( \widehat{\mathbf{F}}_\ell^\top \widehat{\mathbf{Z}}^{(t)} \right) + \widehat{\varepsilon}_\ell^{(t)}, \tag{12}$$

with $\widehat{\varepsilon}_\ell^{(t)} \sim p_{\varepsilon_\ell^{(t)}}$. The latent process $\{\mathbf{Z}^{(t)}\}$ is said to be *identifiable* upto permutation and scaling, if:

$$p(\mathbf{X}^{(t)}(\ell)|\mathbf{Z}^{(t)}; \mathbf{F}_\ell) = p(\widehat{\mathbf{X}}^{(t)}(\ell)|\widehat{\mathbf{Z}}^{(t)}; \widehat{\mathbf{F}}_\ell), \quad \forall \ell \in \mathcal{G}, t \in [T]$$

$$\implies \widehat{\mathbf{Z}}_t = PS\mathbf{Z}^{(t)}, \text{ and } \{F_{\psi_i}\}_{i=1}^{D} = \left\{ F_{\widehat{\psi_i}} \right\}_{i=1}^{D},$$

for some permutation matrix $P$ and scaling matrix $S$.

We first show a useful result that allows us to "denoise" the observation space and enables the point-wise equality of the transformed latent time series.

**Lemma 1** (Denoising Lemma)**.** *Suppose* $\mathbf{X} = \left( \left\{ \mathbf{Z}^{(t)} \right\}_{t=1}^{T}, \{F_{\psi_i}\}_{i=1}^{D}, \{g_\ell \mid \ell \in \mathcal{G}\}, p_{\varepsilon_\ell^{(t)}} \right)$ *and* $\widehat{\mathbf{X}} = \left( \left\{ \widehat{\mathbf{Z}}^{(t)} \right\}_{t=1}^{T}, \left\{ F_{\widehat{\psi_i}} \right\}_{i=1}^{D}, \{\widehat{g}_\ell \mid \ell \in \mathcal{G}\}, p_{\varepsilon_\ell^{(t)}} \right)$ *are two SFPs with observational distributions* $p(\mathbf{X}^{(t)}|\mathbf{Z}^{(t)}; \mathbf{F})$ *and* $p(\widehat{\mathbf{X}}^{(t)}|\widehat{\mathbf{Z}}^{(t)}; \widehat{\mathbf{F}})$ *respectively, such that*

$$\mathbf{X}^{(t)}(\ell) = g_\ell \left( \mathbf{F}_\ell^\top \mathbf{Z}^{(t)} \right) + \varepsilon_\ell^{(t)},$$

*and*

$$\widehat{\mathbf{X}}^{(t)}(\ell) = \widehat{g}_\ell \left( \widehat{\mathbf{F}}_\ell^\top \widehat{\mathbf{Z}}^{(t)} \right) + \widehat{\varepsilon}_\ell^{(t)},$$

*where* $\varepsilon_\ell^{(t)}, \widehat{\varepsilon}_\ell^{(t)} \sim p_{\varepsilon_\ell^{(t)}}$. *Assume that for all* $\ell \in \mathcal{G}$ *and* $t \in [T]$, *the set* $\left\{ x \in \mathbb{R} \mid \varphi_{\varepsilon_\ell^{(t)}}(x) = 0 \right\}$ *has measure zero where* $\varphi_{\varepsilon_\ell^{(t)}}$ *represents the characteristic function of the density* $p_{\varepsilon_\ell^{(t)}}$. *If*

$$p(\mathbf{X}^{(t)}(\ell) = x|\mathbf{Z}^{(t)}; \mathbf{F}_\ell) = p(\widehat{\mathbf{X}}^{(t)}(\ell) = x|\widehat{\mathbf{Z}}^{(t)}; \widehat{\mathbf{F}}_\ell) \quad \forall \ell \in \mathcal{G}, t \in [T]. \tag{13}$$

*Then we have that*

$$g_\ell \left( \mathbf{F}_\ell^\top \mathbf{Z}^{(t)} \right) = \widehat{g}_\ell \left( \widehat{\mathbf{F}}_\ell^\top \widehat{\mathbf{Z}}^{(t)} \right) \quad \forall \ell \in \mathcal{G}, t \in [T].$$

*Proof.* Our argument is similar to Step I of Appendix B.2.2 in Khemakhem et al. (2020).

Note that we can write

$$p(\mathbf{X}^{(t)}(\ell) = x|\mathbf{Z}^{(t)}; \mathbf{F}_\ell) = p_{\varepsilon_\ell^{(t)}}(x - \bar{x}) = \int_{\mathbb{R}} \delta_{\bar{x}}(z) \, p_{\varepsilon_\ell^{(t)}}(x - z) dz = \delta_{\bar{x}} * p_{\varepsilon_\ell^{(t)}}(x), \tag{14}$$

where $\bar{x} = g_\ell \left( \mathbf{F}_\ell^\top \mathbf{Z}^{(t)} \right)$, $\delta_{\bar{x}}$ denotes the Dirac-delta distribution centered at $\bar{x}$ and $*$ denotes the convolution operator. Similarly,

$$p(\widehat{\mathbf{X}}^{(t)}(\ell) = x|\widehat{\mathbf{Z}}^{(t)}; \widehat{\mathbf{F}}_\ell) = p_{\varepsilon_\ell^{(t)}}(x - \tilde{x}) = \delta_{\tilde{x}} * p_{\varepsilon_\ell^{(t)}}(x), \tag{15}$$

where $\tilde{x} = \widehat{g}_\ell \left( \widehat{\mathbf{F}}_\ell^\top \widehat{\mathbf{Z}}^{(t)} \right)$.

From equation 13 and the equations 14 and 15, we obtain

$$\delta_{\bar{x}} * p_{\varepsilon_\ell^{(t)}}(x) = \delta_{\tilde{x}} * p_{\varepsilon_\ell^{(t)}}(x).$$

Taking the Fourier Transform on both sides of the equation, we obtain,

$$e^{is\bar{x}} \varphi_{\varepsilon_\ell^{(t)}}(s) = e^{is\tilde{x}} \varphi_{\varepsilon_\ell^{(t)}}(s).$$

Since $\varphi_{\varepsilon_\ell^{(t)}} \neq 0$ almost everywhere, we have that $e^{is\bar{x}} = e^{is\tilde{x}}$ for almost all values of $s$. This implies that

$$\bar{x} = \tilde{x}, \text{ i.e., } g_\ell\left(\mathbf{F}_\ell^\top \mathbf{Z}^{(t)}\right) = \widehat{g}_\ell\left(\widehat{\mathbf{F}}_\ell^\top \widehat{\mathbf{Z}}^{(t)}\right) \forall \ell \in \mathcal{G}, t \in [T].$$

$\square$

### A.2.1. LINEAR IDENTIFIABILITY

Our first result shows that for the case with no nonlinearity, that is $g_\ell(y) = y$ for all $\ell \in \mathcal{G}$, the latents and spatial factors are identifiable.

**Theorem 1** (Identifiability of Linear SFPs). *Suppose we are given two SFPs* $\mathbf{X} = SFP\left(\mathbf{Z}, \mathcal{F}, \{g_\ell \mid \ell \in \mathcal{G}\}, p_{\varepsilon_\ell^{(t)}}\right)$ *and* $\widehat{\mathbf{X}} = SFP\left(\widehat{\mathbf{Z}}, \widehat{\mathcal{F}}, \{\widehat{g}_\ell \mid \ell \in \mathcal{G}\}, p_{\varepsilon_\ell^{(t)}}\right)$ *specified by Equations 11 and 12 respectively, such that both $\mathcal{F}$ and $\widehat{\mathcal{F}}$ belong to the same (potentially infinite) family of linearly independent functions. Further, suppose that the following conditions are satisfied:*

1. ***Linearity:*** *For all $\ell \in \mathcal{G}$, $g_\ell = \widehat{g}_\ell = Id$, where Id is the identity function.*

2. ***Non-degenerate latent processes:*** *For all $d \in [D]$, $\exists t_0 \in [T]$ such that $\mathbf{Z}_d^{(t_0)} \neq 0$, that is, none of the time series is trivially zero. A similar condition holds for $\widehat{\mathbf{Z}}$.*

3. ***Characteristic function of the noise:*** *For all $\ell \in \mathcal{G}$ and $t \in [T]$, the set $\left\{x \in \mathbb{R} \mid \varphi_{\varepsilon_\ell^{(t)}}(x) = 0\right\}$ has measure zero where $\varphi_{\varepsilon_\ell^{(t)}}$ represents the characteristic function of the density $p_{\varepsilon_\ell^{(t)}}$.*

*If $p(\mathbf{X}^{(t)}(\ell)|\mathbf{Z}^{(t)}; \mathbf{F}_\ell) = p\left(\widehat{\mathbf{X}}^{(t)}(\ell)|\widehat{\mathbf{Z}}^{(t)}; \widehat{\mathbf{F}}_\ell\right)$ for every $\ell \in \mathcal{G}$ and $t \in [T]$, then $\mathbf{Z} = P\widetilde{\mathbf{Z}}$ and $\mathcal{F} = \widehat{\mathcal{F}}$ for some permutation matrix $P$.*

*Proof.* By Lemma 1,

$$p(\mathbf{X}^{(t)}(\ell)|\mathbf{Z}^{(t)}; \mathbf{F}_\ell) = p\left(\widehat{\mathbf{X}}^{(t)}(\ell)|\widehat{\mathbf{Z}}^{(t)}; \widehat{\mathbf{F}}_\ell\right)$$

$$\implies \mathbf{F}_\ell^\top \mathbf{Z}^{(t)} = \widehat{\mathbf{F}}_\ell^\top \widehat{\mathbf{Z}}^{(t)} \quad \forall \ell \in \mathcal{G}, \quad t \in [T]$$

$$\implies \sum_{j=1}^D F_{\psi_j}(\ell)\mathbf{Z}_j^{(t)} = \sum_{j=1}^D F_{\widehat{\psi}_j}(\ell)\widehat{\mathbf{Z}}_j^{(t)} \quad \forall \ell \in \mathcal{G}, \quad t \in [T]$$

$$\implies \sum_{j=1}^D F_{\psi_j}(\ell)\mathbf{Z}_j^{(t)} - \sum_{j=1}^D F_{\widehat{\psi}_j}(\ell)\widehat{\mathbf{Z}}_j^{(t)} = 0 \quad \forall \ell \in \mathcal{G}, \quad t \in [T] \tag{16}$$

Suppose $\{\psi_1, \dots, \psi_D\} \cap \left\{\widehat{\psi}_1, \dots, \widehat{\psi}_D\right\} = \varnothing$. Then, due to the fact that both $\mathcal{F}$ and $\widehat{\mathcal{F}}$ are subsets from the same family of linearly independent functions, this would imply that $\mathbf{Z}_j^{(t)} = \widehat{\mathbf{Z}}_j^{(t)} = 0 \quad \forall j \in [D], t \in [T]$, which is a contradiction since we assume that none of the time series are all 0. This implies that $\{\psi_1, \dots, \psi_D\} \cap \left\{\widehat{\psi}_1, \dots, \widehat{\psi}_D\right\} \neq \varnothing$. Assume $V = \left\{(i,j) : \psi_i = \tilde{\psi}_j\right\}$ and define $I = \{i : \exists j \text{ such that } (i,j) \in V\}$, $J = \{j : \exists i \text{ such that } (i,j) \in V\}$. Define the function $\mathcal{V} : I \to J, \quad \mathcal{V}(i) = j$ such that $(i,j) \in V$. Then equation 16 can be written as:

$$\sum_{\substack{j=1 \\ j \notin I}}^D F_{\psi_j}(\ell)\mathbf{Z}_j^{(t)} - \sum_{\substack{j=1 \\ j \notin J}}^D F_{\widehat{\psi}_j}(\ell)\widehat{\mathbf{Z}}_j^{(t)} + \sum_{\substack{j=1 \\ j \in I}}^D F_{\psi_j}(\ell)\left(\mathbf{Z}_j^{(t)} - \widehat{\mathbf{Z}}_{\mathcal{V}(j)}^{(t)}\right) = 0 \quad \forall \ell \in \mathcal{G}, t \in [T].$$

If $I^{\complement} \neq \varnothing$, then $\mathbf{Z}_j^{(t)} = 0 \ \forall j \in I^{\complement}$ due to the linear independence of $F_{\psi_j}$, which contradicts our assumption of non-zero time series. Therefore, we must have that $I^{\complement} = \varnothing$, which implies that $\{\psi_1, \ldots, \psi_D\} = \{\tilde{\psi}_1, \ldots, \tilde{\psi}_D\}$, and $\mathbf{Z}_j^{(t)} = \mathbf{Z}_{\mathcal{V}(j)}^{(t)} \ \forall j \in [D], t \in [T]$. $\qquad\square$

### A.2.2. GENERAL IDENTIFIABILITY

We now turn our attention to the general setting where $g_\ell$ can be nonlinear. Before proving the identifiability result, we state several useful lemmas from real analysis. We first recall some useful properties of real analytic functions.

**Definition 3** (Real Analytic Functions)**.** Let $U$ be an open set in $\mathbb{R}^K$. A function $f : U \to \mathbb{R}$ is real analytic (or simply analytic) if at each point $x \in U$, the function $f$ has a convergent power series representation that converges (absolutely) to $f(x)$ in some neighborhood of $x$.

In other words, real analytic functions can be written using a power series representation for every point in the domain. Some examples include polynomial functions and the family of RBF kernels. Next, we recall the following, well-known identity theorem for real analytic functions.

**Lemma 2** (Identity Theorem for Real Analytic Functions (Lebl, 2022))**.** *Suppose* $f : U \to \mathbb{R}$ *is a real analytic function defined on a connected domain* $U \subset \mathbb{R}^K$. *If* $f = 0$ *on a nonempty, open subset* $V \subset U$, *then* $f \equiv 0$ *on* $U$.

We also recall another important result about real analytic functions which we will use in our proof.

**Lemma 3** (Zero sets of analytic functions have zero measure (Mityagin, 2015))**.** *Let* $f : U \to \mathbb{R}$ *be a real analytic function on a connected open domain* $U \subset \mathbb{R}^K$. *If* $f$ *is not identically zero, then its zero set*
$$\Lambda_f = \{x \in U \mid f(x) = 0\}$$
*has a zero Lebesgue measure* $\mu(\Lambda_f) = 0$.

Using these results, we prove some useful results about linearly independent families of real analytic functions.

**Lemma 4.** *Suppose* $F = \{f_1, \ldots, f_D\}$ *and* $G = \{g_1, \ldots, g_D\}$ *are two sets of linearly independent, real analytic functions defined on* $\mathcal{G} = (0, 1)^K$. *Define the matrices*

$$\mathcal{M}_F(\ell_1, \ldots, \ell_D) = \begin{bmatrix} f_1(\ell_1) & \cdots & f_1(\ell_D) \\ \vdots & & \vdots \\ f_D(\ell_1) & \cdots & f_D(\ell_D) \end{bmatrix} \qquad \mathcal{M}_G(\ell_1, \ldots, \ell_D) = \begin{bmatrix} g_1(\ell_1) & \cdots & g_1(\ell_D) \\ \vdots & & \vdots \\ g_D(\ell_1) & \cdots & g_D(\ell_D) \end{bmatrix}. \qquad (17)$$

*Let*
$$\Phi_F = \{\boldsymbol{\ell} = (\ell_1, \ldots, \ell_D) \mid \ell_i \in \mathcal{G}, \mathcal{M}_F(\boldsymbol{\ell}) \text{ is full rank}\} \qquad (18)$$
*denote the set of* $D-$*tuples in* $\mathbb{R}^K$ *for which the matrix* $\mathcal{M}_F$ *is full rank. Then*

1. $\mu(\Phi_F \cap \Phi_G) = 1$. *In particular,* $\Phi_F \cap \Phi_G \neq \varnothing$.

2. $\Phi_F \cap \Phi_G$ *is an open set.*

*Proof.* 1. The complement of the set $\Phi_F$ is
$$\Phi_F^{\complement} = \{\boldsymbol{\ell} = (\ell_1, \ldots, \ell_D) \mid \ell_i \in \mathcal{G}, \det(\mathcal{M}_F)(\boldsymbol{\ell}) = 0\}.$$
Since $\det(\mathcal{M}_F)(\boldsymbol{\ell})$ is a polynomial in real analytic functions, it is also a real analytic function in $\mathbb{R}^{DK}$.

Note that since $F$ is linearly independent, $\Phi_F$ is non-empty. [1] Thus, $\det(\mathcal{M}_F)(\boldsymbol{\ell})$ is not identically zero. By Lemma 3,

---

[1]See for example:
https://math.stackexchange.com/questions/3516189/prove-existence-of-evaluation-points-such-that-the-matrix-has-nonzero-determinan.

$\mu\left(\Phi_F^{\complement}\right) = 0$. Similarly, $\mu\left(\Phi_G^{\complement}\right) = 0$, and $\mu\left(\Phi_F^{\complement} \cap \Phi_G^{\complement}\right) \leq \mu\left(\Phi_F^{\complement}\right) = 0 \implies \mu\left(\Phi_F^{\complement} \cap \Phi_G^{\complement}\right) = 0$. Thus, we obtain that

$$\mu\left(\Phi_F \cap \Phi_G\right) = \mu\left(\Phi_F\right) + \mu\left(\Phi_G\right) - \mu\left(\Phi_F \cup \Phi_G\right)$$
$$= \mu(\mathcal{G}) - \mu\left(\Phi_F^{\complement}\right) + \mu(\mathcal{G}) - \mu\left(\Phi_G^{\complement}\right) - \left(\mu(\mathcal{G}) - \mu\left(\Phi_F^{\complement} \cap \Phi_G^{\complement}\right)\right)$$
$$= \mu(\mathcal{G}) - \mu\left(\Phi_F^{\complement}\right) - \mu\left(\Phi_G^{\complement}\right) + \mu\left(\Phi_F^{\complement} \cap \Phi_G^{\complement}\right)$$
$$= 1.$$

2. Since the function $\det\left(\mathcal{M}_F\right)(\ell)$ is real analytic, it is also continuous. Since the zero set of a continuous function is closed, $\Phi_F^{\complement}$ is closed, which implies that $\Phi_F$ is open. Similarly, $\Phi_G$ is open. Since the finite intersection of open sets is open, $\Phi_F \cap \Phi_G$ is open. $\qquad\square$

**Lemma 5.** *Suppose $F = \{f_1, \ldots, f_D\}$ is a set of linearly independent real analytic functions defined on $\mathcal{G} = (0,1)^K$. Then $F$ is linearly independent in every non-empty open set $U \subset \mathcal{G}$.*

*Proof.* Suppose there exists a nonempty open set $U \subset \mathcal{G}$ in which $F$ is linearly dependent. This implies $\exists c_1, \ldots c_D \in \mathbb{R}$, not all zero, such that

$$\sum_{i=1}^{D} c_i f_i(\ell) = 0 \qquad \forall \ell \in U.$$

Then, by Lemma 2, the real analytic function $\sum_{i=1}^{D} c_i f_i(\ell)$ is identically zero everywhere in $\mathcal{G}$, which is a contradiction to the linear independence of $F$. $\qquad\square$

We are now ready to prove the identifiability of the model under general, diffeomorphic non-linearities in the mapping between the latent and observational space.

**Theorem 2** (Identifiability of General SFPs). *Suppose we are given two SFPs $\mathbf{X} = SFP\left(\mathbf{Z}, \mathcal{F}, \{g_\ell \mid \ell \in \mathcal{G}\}, p_{\varepsilon_\ell^{(t)}}\right)$ and $\widehat{\mathbf{X}} = SFP\left(\widehat{\mathbf{Z}}, \widehat{\mathcal{F}}, \{\widehat{g}_\ell \mid \ell \in \mathcal{G}\}, p_{\varepsilon_\ell^{(t)}}\right)$ specified by Equations 11 and 12 respectively, such that both $\mathcal{F}$ and $\widehat{\mathcal{F}}$ belong to the same (potentially infinite) family of linearly independent functions. Further, suppose that the following conditions are satisfied:*

1. *__Diffeomorphisms:__ For all $\ell \in \mathcal{G}$, $g_\ell, \widehat{g}_\ell$ are diffeomorphisms, that is, $g_\ell, \widehat{g}_\ell$ are invertible and continuously differentiable, and their inverses are also continuously differentiable.*

2. *__Real analytic functions:__ All the functions $F_{\psi_i} \in \mathcal{F}, F_{\widehat{\psi}_i} \in \widehat{\mathcal{F}}$ are real analytic.*

3. *__Non-degenerate latent processes:__ For all $d \in [D]$, $\exists\, t_0 \in [T]$ such that $\mathbf{Z}_d^{(t_0)} \neq 0$, that is, none of the time series is trivially zero. A similar condition holds for $\widehat{\mathbf{Z}}$.*

4. *__Characteristic function of the noise:__ For all $\ell \in \mathcal{G}$ and $t \in [T]$, the set $\left\{x \in \mathbb{R} \mid \varphi_{\varepsilon_\ell^{(t)}}(x) = 0\right\}$ has measure zero where $\varphi_{\varepsilon_\ell^{(t)}}$ represents the characteristic function of the density $p_{\varepsilon_\ell^{(t)}}$.*

*If $p(\mathbf{X}^{(t)}(\ell)|\mathbf{Z}^{(t)}; \mathbf{F}_\ell) = p\left(\widehat{\mathbf{X}}^{(t)}(\ell)|\widehat{\mathbf{Z}}^{(t)}; \widehat{\mathbf{F}}_\ell\right)$ for every $\ell \in \mathcal{G}$ and $t \in [T]$, then $\mathbf{Z} = PS\widetilde{\mathbf{Z}}$ and $\mathcal{F} = \widehat{\mathcal{F}}$ for some permutation matrix $P$ and scaling matrix $S$.*

*Proof.* By Lemma 1, we obtain that,

$$p(\mathbf{X}^{(t)}(\ell)|\mathbf{Z}^{(t)}; \mathbf{F}_\ell) = p\left(\widehat{\mathbf{X}}^{(t)}(\ell)|\widehat{\mathbf{Z}}^{(t)}; \widehat{\mathbf{F}}_\ell\right)$$
$$\implies g_\ell\left(\mathbf{F}_\ell^{\top}\mathbf{Z}^{(t)}\right) = \widehat{g}_\ell\left(\widehat{\mathbf{F}}_\ell^{\top}\widehat{\mathbf{Z}}^{(t)}\right) \quad \forall \ell \in \mathcal{G}, \quad t \in [T]$$
$$\implies \mathbf{F}_\ell^{\top}\mathbf{Z}^{(t)} = g_\ell^{-1} \circ \widehat{g}_\ell\left(\widehat{\mathbf{F}}_\ell^{\top}\widehat{\mathbf{Z}}^{(t)}\right) = h_\ell\left(\widehat{\mathbf{F}}_\ell^{\top}\widehat{\mathbf{Z}}^{(t)}\right) \quad \forall \ell \in \mathcal{G}, \quad t \in [T] \qquad (19)$$

where $h_\ell = g_\ell^{-1} \circ \widehat{g}_\ell$. Defining $\Phi_{\mathcal{F}}$ and $\Phi_{\widehat{\mathcal{F}}}$ as in equation 17, using Lemma 4, we get that $\Phi_{\mathcal{F}} \cap \Phi_{\widehat{\mathcal{F}}} \neq \varnothing$. Pick an arbitrary point $\boldsymbol{\ell} = (\ell_1, \ldots, \ell_D) \in \Phi_{\mathcal{F}} \cap \Phi_{\widehat{\mathcal{F}}}$. Then, the matrices

$$\mathcal{M}_{\mathcal{F}}(\boldsymbol{\ell}) = \begin{bmatrix} F_{\psi_1}(\ell_1) & \ldots & F_{\psi_1}(\ell_D) \\ \vdots & & \vdots \\ F_{\psi_D}(\ell_1) & \ldots & F_{\psi_D}(\ell_D) \end{bmatrix} \qquad \mathcal{M}_{\widehat{\mathcal{F}}}(\boldsymbol{\ell}) = \begin{bmatrix} F_{\widehat{\psi}_1}(\ell_1) & \ldots & F_{\widehat{\psi}_1}(\ell_D) \\ \vdots & & \vdots \\ F_{\widehat{\psi}_D}(\ell_1) & \ldots & F_{\widehat{\psi}_D}(\ell_D) \end{bmatrix}$$

are invertible. Evaluating equation 19 at the $D$ points $\ell_1, \ldots, \ell_D$, we obtain:

$$\mathcal{M}_{\mathcal{F}}(\boldsymbol{\ell})^\top \mathbf{Z}^{(t)} = \mathscr{H}_{\boldsymbol{\ell}}\left(\mathcal{M}_{\widehat{\mathcal{F}}}(\boldsymbol{\ell})^\top \widehat{\mathbf{Z}}^{(t)}\right)$$

$$\implies \mathbf{Z}^{(t)} = \left(\mathcal{M}_{\mathcal{F}}(\boldsymbol{\ell})^\top\right)^{-1} \mathscr{H}_{\boldsymbol{\ell}}\left(\mathcal{M}_{\widehat{\mathcal{F}}}(\boldsymbol{\ell})^\top \widehat{\mathbf{Z}}^{(t)}\right) := \Theta_{\boldsymbol{\ell}}\left(\widehat{\mathbf{Z}}^{(t)}\right), \tag{20}$$

where $\mathscr{H}_{\boldsymbol{\ell}}(y) = \left[h_{\ell_1}(y_1), \ldots, h_{\ell_D}(y_D)\right]^\top$ for $y \in \mathbb{R}^D$. Since $\mathscr{H}_{\boldsymbol{\ell}}$ is a component-wise invertible function, and $\mathcal{M}_{\mathcal{F}}(\boldsymbol{\ell}), \mathcal{M}_{\widehat{\mathcal{F}}}(\boldsymbol{\ell})$ are invertible matrices, the map $\Theta_{\boldsymbol{\ell}}$ is an invertible map between $\mathbf{Z}^{(t)}$ and $\widehat{\mathbf{Z}}^{(t)}$. Furthermore, the above argument can be repeated for any arbitrary $\boldsymbol{\ell}' \in \Phi_{\mathcal{F}} \cap \Phi_{\widehat{\mathcal{F}}}$ to obtain $\mathbf{Z}^{(t)} = \Theta_{\boldsymbol{\ell}'}\left(\widehat{\mathbf{Z}}^{(t)}\right)$. Thus, we have that

$$\mathbf{Z}^{(t)} = \Theta_{\boldsymbol{\ell}}\left(\widehat{\mathbf{Z}}^{(t)}\right) = \Theta_{\boldsymbol{\ell}'}\left(\widehat{\mathbf{Z}}^{(t)}\right) \text{ for } \boldsymbol{\ell}, \boldsymbol{\ell}' \in \Phi_{\mathcal{F}} \cap \Phi_{\widehat{\mathcal{F}}}. \tag{21}$$

Now, consider the Jacobian $\dfrac{\partial \mathbf{Z}^{(t)}}{\partial \widehat{\mathbf{Z}}^{(t)}} = J_{\Theta_{\boldsymbol{\ell}}}\left(\widehat{\mathbf{Z}}^{(t)}\right)$ of the transformation $\Theta_{\boldsymbol{\ell}}$. Our goal is to prove that $J_{\Theta_{\boldsymbol{\ell}}}$ is a permutation scaling matrix, that is, for each $i \in [D]$, $\dfrac{\partial \mathbf{Z}_i^{(t)}}{\partial \widehat{\mathbf{Z}}_j^{(t)}}$ is non-zero for exactly one value of $j \in [D]$. By the chain rule,

$$J_{\Theta_{\boldsymbol{\ell}}}\left(\widehat{\mathbf{Z}}^{(t)}\right) = \left(\mathcal{M}_{\mathcal{F}}(\boldsymbol{\ell})^\top\right)^{-1} \mathscr{H}_{\boldsymbol{\ell}}'\left(\mathcal{M}_{\widehat{\mathcal{F}}}(\boldsymbol{\ell})^\top \widehat{\mathbf{Z}}^{(t)}\right) \mathcal{M}_{\widehat{\mathcal{F}}}(\boldsymbol{\ell})^\top \tag{22}$$

where $\mathscr{H}_{\boldsymbol{\ell}}'(y) = \begin{bmatrix} h_{\ell_1}'(y_1) & & \\ & \ddots & \\ & & h_{\ell_D}'(y_d) \end{bmatrix}$ for $y \in \mathbb{R}^D$.

Taking the log-determinant of equation 22, we obtain

$$\log\left|\det J_{\Theta_{\boldsymbol{\ell}}}\left(\widehat{\mathbf{Z}}^{(t)}\right)\right| = \log\left|\det\left(\mathcal{M}_{\mathcal{F}}(\boldsymbol{\ell})^\top\right)^{-1}\right| + \log\left|\det\left(\mathcal{M}_{\widehat{\mathcal{F}}}(\boldsymbol{\ell})^\top\right)\right| + \sum_{k=1}^{D} \log\left|h_{\ell_k}'\left(\widehat{\mathbf{F}}_{\ell_k} \widehat{\mathbf{Z}}^{(t)}\right)\right| \tag{23}$$

Differentiating both sides of equation 23 with respect to $\widehat{\mathbf{Z}}_i^{(t)}$, we obtain:

$$\frac{\partial}{\partial \widehat{\mathbf{Z}}_i^{(t)}} \log\left|\det J_{\Theta_{\boldsymbol{\ell}}}\left(\widehat{\mathbf{Z}}^{(t)}\right)\right| = \sum_{k=1}^{D} \frac{h_{\ell_k}''\left(\widehat{\mathbf{F}}_{\ell_k}^\top \widehat{\mathbf{Z}}^{(t)}\right)}{h_{\ell_k}'\left(\widehat{\mathbf{F}}_{\ell_k}^\top \widehat{\mathbf{Z}}^{(t)}\right)} F_{\widehat{\psi}_i}(\ell_k). \tag{24}$$

By the second part of Lemma 4, $\Phi_{\mathcal{F}} \cap \Phi_{\widehat{\mathcal{F}}}$ is open. Thus, $\exists r > 0$ such that $B_{DK}\left(\boldsymbol{\ell}, r\right) \subset \Phi_{\mathcal{F}} \cap \Phi_{\widehat{\mathcal{F}}}$, where $B_{DK}(\boldsymbol{\ell}, r)$ denotes the open ball in $\mathbb{R}^{DK}$ centered around $\boldsymbol{\ell}$ with radius $r > 0$. Choose an arbitrary unit vector $\mathbf{u} \in \mathbb{R}^K$ and consider the point $\boldsymbol{\ell}' = (\ell_1 + \delta \mathbf{u}, \ell_2, \ldots, \ell_D)$ such that $\delta < r$. Since $J_{\Theta_{\boldsymbol{\ell}}}\left(\widehat{\mathbf{Z}}^{(t)}\right) = \dfrac{\partial \mathbf{Z}^{(t)}}{\partial \widehat{\mathbf{Z}}^{(t)}} = J_{\Theta_{\boldsymbol{\ell}'}}\left(\widehat{\mathbf{Z}}^{(t)}\right)$ for $\boldsymbol{\ell}, \boldsymbol{\ell}' \in \Phi_{\mathcal{F}} \cap \Phi_{\widehat{\mathcal{F}}}$,

$\log \left| \det J_{\Theta_{\boldsymbol{\ell}}} \left( \widehat{\mathbf{Z}}^{(t)} \right) \right| = \log \left| \det J_{\Theta_{\boldsymbol{\ell}'}} \left( \widehat{\mathbf{Z}}^{(t)} \right) \right|$. From equation 24, we obtain

$$\frac{\partial}{\partial \widehat{\mathbf{Z}}_i^{(t)}} \log \left| \det J_{\Theta_{\boldsymbol{\ell}}} \left( \widehat{\mathbf{Z}}^{(t)} \right) \right| = \sum_{k=1}^{D} \frac{h_{\ell_k}'' \left( \widehat{\mathbf{F}}_{\ell_k}^{\top} \widehat{\mathbf{Z}}^{(t)} \right)}{h_{\ell_k}' \left( \widehat{\mathbf{F}}_{\ell_k}^{\top} \widehat{\mathbf{Z}}^{(t)} \right)} F_{\widehat{\psi}_i}(\ell_k)$$

$$= \frac{h_{\ell_1 + \delta \mathbf{u}}'' \left( \widehat{\mathbf{F}}_{\ell_1 + \delta \mathbf{u}}^{\top} \widehat{\mathbf{Z}}^{(t)} \right)}{h_{\ell_1 + \delta \mathbf{u}}' \left( \widehat{\mathbf{F}}_{\ell_1 + \delta \mathbf{u}}^{\top} \widehat{\mathbf{Z}}^{(t)} \right)} F_{\widehat{\psi}_i}(\ell_1 + \delta \mathbf{u}) + \sum_{k=2}^{D} \frac{h_{\ell_k}'' \left( \widehat{\mathbf{F}}_{\ell_k}^{\top} \widehat{\mathbf{Z}}^{(t)} \right)}{h_{\ell_k}' \left( \widehat{\mathbf{F}}_{\ell_k}^{\top} \widehat{\mathbf{Z}}^{(t)} \right)} F_{\widehat{\psi}_i}(\ell_k)$$

$$= \frac{\partial}{\partial \widehat{\mathbf{Z}}_i^{(t)}} \log \left| \det J_{\Theta_{\boldsymbol{\ell}'}} \left( \widehat{\mathbf{Z}}^{(t)} \right) \right|$$

$$\implies \frac{h_{\ell_1 + \delta \mathbf{u}}'' \left( \widehat{\mathbf{F}}_{\ell_1 + \delta \mathbf{u}}^{\top} \widehat{\mathbf{Z}}^{(t)} \right)}{h_{\ell_1 + \delta \mathbf{u}}' \left( \widehat{\mathbf{F}}_{\ell_1 + \delta \mathbf{u}}^{\top} \widehat{\mathbf{Z}}^{(t)} \right)} F_{\widehat{\psi}_i}(\ell_1 + \delta \mathbf{u}) = \frac{h_{\ell_1}'' \left( \widehat{\mathbf{F}}_{\ell_1}^{\top} \widehat{\mathbf{Z}}^{(t)} \right)}{h_{\ell_1}' \left( \widehat{\mathbf{F}}_{\ell_1}^{\top} \widehat{\mathbf{Z}}^{(t)} \right)} F_{\widehat{\psi}_i}(\ell_1) \tag{25}$$

Since $\delta < r$ and $\mathbf{u}$ are arbitrary, the function

$$\Gamma_i(\widehat{\mathbf{Z}}^{(t)}) = \frac{h_\ell'' \left( \widehat{\mathbf{F}}_\ell^{\top} \widehat{\mathbf{Z}}^{(t)} \right)}{h_\ell' \left( \widehat{\mathbf{F}}_\ell^{\top} \widehat{\mathbf{Z}}^{(t)} \right)} F_{\widehat{\psi}_i}(\ell) \tag{26}$$

is a constant with respect to $\ell$ for $\ell \in B_K(\ell_1, r)$. Differentiating equation 23 with respect to $\widehat{\mathbf{Z}}_j^{(t)}, j \neq i$ and repeating the above argument, we obtain that

$$\Gamma_j(\widehat{\mathbf{Z}}^{(t)}) = \frac{h_\ell'' \left( \widehat{\mathbf{F}}_\ell^{\top} \widehat{\mathbf{Z}}^{(t)} \right)}{h_\ell' \left( \widehat{\mathbf{F}}_\ell^{\top} \widehat{\mathbf{Z}}^{(t)} \right)} F_{\widehat{\psi}_j}(\ell) \tag{27}$$

is constant with respect to $\ell$ for $\ell \in B_K(\ell_1, r)$. From equations 26 and 27,

$$F_{\widehat{\psi}_i}(\ell) \Gamma_j(\widehat{\mathbf{Z}}^{(t)}) = F_{\widehat{\psi}_j}(\ell) \Gamma_i(\widehat{\mathbf{Z}}^{(t)}), \quad \ell \in B_K(\ell_1, r). \tag{28}$$

Note that since $F_{\widehat{\psi}_i}$ and $F_{\widehat{\psi}_j}$ are linearly independent in $\mathcal{G}$, by Lemma 5, they are also linearly independent for $\ell \in B_K(\ell_1, r)$, which implies that $\Gamma_i(\widehat{\mathbf{Z}}^{(t)}) = \Gamma_j(\widehat{\mathbf{Z}}^{(t)}) = 0$. Since these arguments can be repeated for any distinct indices $i, j \in [D]$, we infer that

$$\Gamma_i(\widehat{\mathbf{Z}}^{(t)}) = \frac{h_\ell'' \left( \widehat{\mathbf{F}}_\ell^{\top} \widehat{\mathbf{Z}}^{(t)} \right)}{h_\ell' \left( \widehat{\mathbf{F}}_\ell^{\top} \widehat{\mathbf{Z}}^{(t)} \right)} F_{\widehat{\psi}_i}(\ell) = 0 \qquad \forall i \in [D], \ell \in B_K(\ell_1, r). \tag{29}$$

Note that equation 29 implies that $h_\ell'' \left( \widehat{\mathbf{F}}_\ell^{\top} \widehat{\mathbf{Z}}^{(t)} \right) \equiv 0 \quad \forall \ell \in B_K(\ell_1, r), \widehat{\mathbf{Z}}^{(t)} \in \mathbb{R}^D$, otherwise, if for some $\ell_0$ and $\widehat{\mathbf{Z}}^{(t)}$, the value of $h_{\ell_0}'' \left( \widehat{\mathbf{F}}_{\ell_0}^{\top} \widehat{\mathbf{Z}}^{(t)} \right) \neq 0$, then $F_{\widehat{\psi}_i}(\ell_0) = 0 \quad \forall i \in [D]$ which would contradict the invertibility of $\mathcal{M}_{\widehat{\mathcal{F}}}(\ell_0)$ for $\ell_0 \in B_K(\ell_1, r) \subset \Phi_{\mathcal{F}} \cap \Phi_{\widehat{\mathcal{F}}}$.

Therefore,

$$h_\ell' \left( \widehat{\mathbf{F}}_\ell^{\top} \widehat{\mathbf{Z}}^{(t)} \right) = c_0 \quad \forall \ell \in B_K(\ell_1, r) \tag{30}$$

for some constant $c_0 \in \mathbb{R}$.

Now, we differentiate both sides of equation 19 with respect to $\widehat{\mathbf{Z}}_i^{(t)}$ to obtain:

$$\sum_{k=1}^{D} F_{\psi_k}(\ell) \frac{\partial \mathbf{Z}_k^{(t)}}{\partial \widehat{\mathbf{Z}}_i^{(t)}} = h_\ell' \left( \widehat{\mathbf{F}}_\ell^{\top} \widehat{\mathbf{Z}}^{(t)} \right) F_{\widehat{\psi}_i}(\ell). \tag{31}$$

For $\ell \in B_K(\ell_1, r)$, the above equation becomes:

$$\sum_{k=1}^{D} F_{\psi_k}(\ell) \frac{\partial \mathbf{Z}_k^{(t)}}{\partial \widehat{\mathbf{Z}}_i^{(t)}} = c_0 F_{\widehat{\psi}_i}(\ell). \tag{32}$$

Once again, using Lemma 5, $\mathcal{F} = \{F_{\psi_1}, \ldots, F_{\psi_D}\}$ is linearly independent for $\ell \in B_K(\ell_1, r)$. If $F_{\widehat{\psi}_i} \neq F_{\psi_k}$ for some

$k \in [D]$, then due to linear independence, $\frac{\partial \mathbf{Z}_k^{(t)}}{\partial \widehat{\mathbf{Z}}_i^{(t)}} = 0 \quad \forall k \in [D]$. However, this leads to a contradiction, since $\mathbf{Z}^{(t)}$ and $\widehat{\mathbf{Z}}^{(t)}$ are related by an invertible map, which means the corresponding Jacobian matrix is full-rank.

Therefore, $F_{\widehat{\psi}_i} = F_{\psi_k}$ for some $k \in [D]$, and $\frac{\partial \mathbf{Z}_k^{(t)}}{\partial \widehat{\mathbf{Z}}_i^{(t)}}$ is non-zero for exactly one $k \in [D]$. Repeating the argument for all $i \in [D]$ yields the result. $\qquad\square$

### A.3. Theoretical assumptions for Rhino

Since our method, SPACY, relies on Rhino for latent causal discovery, we list the theoretical assumptions used in Gong et al. (2023) for the sake of completeness.

The Rhino model is specified by the following equation:
$$\mathbf{Z}_i^{(t)} = f_i \left( \text{Pa}_{\mathbf{G}}^i(<t), \text{Pa}_{\mathbf{G}}^i(t) \right) + g_i \left( \text{Pa}_{\mathbf{G}}^i(<t), \epsilon_i^{(t)} \right)$$

where $f_i$ is a general differentiable non-linear function, and $g_i$ is a differentiable transform that models the history-dependent noise. The model is known to be identifiable when the following assumptions are satisfied:

**Assumption 1** (Causal Stationarity). (Runge, 2018) The time series $\mathbf{Z}$ with a graph $\mathbf{G}$ is called causally stationary over a time index set $\mathcal{T}$ if and only if for all links $\mathbf{Z}_i^{(t-\tau)} \to \mathbf{Z}_j^{(t)}$ in the graph
$$\mathbf{Z}_i^{(t-\tau)} \not\perp\!\!\!\perp \mathbf{Z}_j^{(t)} \mid \mathbf{Z}^{(t)} \backslash \left\{ \mathbf{Z}_i^{(t-\tau)} \right\} \qquad \text{holds for all } t \in \mathcal{T}.$$
Informally, this assumption states that the causal graph does not change over time, i.e., the resulting time series is stationary.

**Assumption 2** (Causal Markov Property). (Peters et al., 2017) Given a DAG $\mathbf{G}$ and a probability distribution $p$, $p$ is said to satisfy the causal Markov property, if it factorizes according to $\mathbf{G}$, i.e. $p(\mathbf{Z}) = \prod_{i=1}^{D} p\left( \mathbf{Z}_i \mid \text{Pa}_{\mathbf{G}}^i(\mathbf{Z}_i) \right)$. In other words, each variable is independent of its non-descendent given its parents.

**Assumption 3** (Causal Minimality). Given a DAG $\mathbf{G}$ and a probability distribution $p$, $p$ is said to satisfy the causal minimality with respect to $\mathbf{G}$, if $p$ is Markovian with respect to $\mathbf{G}$ but not to any proper subgraph of $\mathbf{G}$.

**Assumption 4** (Causal Sufficiency). A set of observed variables $V$ is said to be causally sufficient for a process $\mathbf{Z}^{(t)}$ if, in the process, every common cause of two or more variables in $V$ is also in $V$, or is constant for all units in the population. In other words, causal sufficiency implies the absence of latent confounders in the data.

**Assumption 5** (Well-defined Density). The likelihood of $\mathbf{Z}^{(t)}$ is absolutely continuous with respect to a Lebesgue or counting measure and $\left| \log p\left( \mathbf{Z}^{(0:T)}; \mathbf{G} \right) \right| < \infty$ for all possible $\mathbf{G}$.

**Assumption 6** (Conditions on $f$ and $g$). The following conditions are satisfied:

1. All functions $f_i, g_i$ and induced distributions are third-order differentiable.

2. $f_i$ is non-linear and not invertible w.r.t any nodes in $\text{Pa}_{\mathbf{G}}^i(t)$.

3. The double derivative $\left( \log p_{g_i} \left( g_i \left( \epsilon_i^{(t)} \mid \text{Pa}_{\mathbf{G}}(<t) \right) \right) \right)''$ w.r.t $\epsilon_i^{(t)}$ is zero at most on some discrete points.

## B. Implementation Details

### B.1. Linear variant of SPACY

We also implement a linear variant of SPACY, called **SPACY-L**. This variant models linear relationships with independent noise. $f_d$ is defined as:
$$f_d \left( \text{Pa}_{\mathbf{G}}^d(\leq t) \right) = \sum_{k=0}^{\tau} \sum_{d'=1}^{D} (\mathbf{G} \circ W)_{d',d}^k \times \mathbf{Z}_{d'}^{t-k}, \tag{33}$$

where $\circ$ denotes the Hadamard product, and $W \in \mathbb{R}^{(\tau+1) \times D \times D}$ is a learned weight tensor. We assume that $\eta_d^t$ is isotropic Gaussian noise.

## B.2. Loss Terms

We explain how we implement the various loss terms in equation 4.

The first term $\log p_\theta \left( \mathbf{X}^{(1:T),n} | \mathbf{Z}^{(1:T),n}, \mathbf{F} \right)$ in equation 4 represents the conditional likelihood of the observed data $\mathbf{X}^{(1:T),n}$ conditioned on $\mathbf{Z}^{(1:T),n}$ and $\mathbf{F}$. This is calculated as the mean squared error (MSE) between the recovered and original time series:

$$\log p_\theta \left( \mathbf{X}^{(1:T),n} | \mathbf{Z}^{(1:T),n}, \mathbf{F} \right) = \sum_{\ell=1}^{L} \left\| \mathbf{X}_\ell^{(1:T),n} - \widehat{\mathbf{X}}_\ell^{(1:T),n} \right\|^2$$

where $\widehat{\mathbf{X}}_\ell^{(t),n} = g_\ell \left( [\mathbf{F}\mathbf{Z}]_\ell^{(t)} \right)$ is the reconstructed time-series from the spatial factor $\mathbf{F}$ and latent time series $\mathbf{Z}$ sampled from the variational distributions.

The term $\log p_\theta \left( \mathbf{Z}^{(1:T),n} | \mathbf{G} \right)$ denotes the conditional likelihood of the latent time-series given the sampled graph $\mathbf{G}$.

For SPACY-L, this is implemented as follows:

$$
\begin{aligned}
\log p_\theta &\left( \mathbf{Z}^{(1:T),n} \Big| \mathbf{G} \right) \\
&= \sum_{t=L}^{T} \sum_{d=1}^{D} \log p_\theta \left( \mathbf{Z}_d^{(t),n} \Big| \mathrm{Pa}_\mathbf{G}^d(\leq t) \right) \\
&= \sum_{t=L}^{T} \sum_{d=1}^{D} \left( \mathbf{Z}_d^{(t),n} - \sum_{k=0}^{\tau} \sum_{j=1}^{D} (\mathbf{G} \circ W)_{j,d}^k \times \mathbf{Z}_j^{(t-k),n} \right)^2 .
\end{aligned}
\tag{34}
$$

In (the nonlinear variant of) SPACY, we use the conditional spline flow model employed in Durkan et al. (2019); Gong et al. (2023). The conditional spline flow model handles more flexible noise distributions, and can also model history-dependent noise. The structural equations are modeled as follows:

$$\mathbf{Z}_d^{(t)} = f_d \left( \mathrm{Pa}_\mathbf{G}^d(< t), \mathrm{Pa}_\mathbf{G}^d(t) \right) + w_d \left( \mathrm{Pa}_\mathbf{G}^d(< t) \right),$$

where $f_d \left( \mathrm{Pa}_\mathbf{G}^d(< t), \mathrm{Pa}_\mathbf{G}^d(t) \right)$ takes the form presented in equation 2. The spline flow model uses a hypernetwork that predicts parameters for the conditional spline flow model, with embeddings $\mathscr{F}$, and hypernetworks $\xi_\eta$ and $\lambda_\eta$. The only difference is that the output dimension of $\xi_\eta$ is different, being equal to the number of spline parameters.

The noise variables $\eta_d^{(t)}$ are described using a conditional spline flow model,

$$p_{w_d}(w_d(\eta_d^{(t)}) \mid \mathrm{Pa}_\mathbf{G}^d(< t)) = p_\eta(\eta_d^{(t)}) \left| \frac{\partial (w_d)^{-1}}{\partial \eta_d^{(t)}} \right|,
\tag{35}$$

with $\eta_d^{(t)}$ modeled as independent Gaussian noise.

The marginal likelihood becomes:

$$
\begin{aligned}
\log p_\theta \left( \mathbf{Z}^{(1:T),n} \Big| \mathbf{G} \right) &= \sum_{t=\tau}^{T} \sum_{d=1}^{D} \log p_\theta \left( \mathbf{Z}_d^{(t),n} \Big| \mathrm{Pa}_\mathbf{G}^d(< t), \mathrm{Pa}_\mathbf{G}^d(t) \right) \\
&= \sum_{t=\tau}^{T} \sum_{d=1}^{D} \log p_{w_d} \left( u_d^{(t),n} \Big| \mathrm{Pa}_\mathbf{G}^d(< t) \right)
\end{aligned}
\tag{36}
$$

where $u_d^{(t),n} = \mathbf{Z}_d^{(t),n} - f_d \left( \mathrm{Pa}_\mathbf{G}^d(< t), \mathrm{Pa}_\mathbf{G}^d(t) \right)$.

The prior distribution $p(\mathbf{G})$ is modeled as follows:

$$p(\mathbf{G}) \propto \exp\left(-\alpha \left\|\mathbf{G}^{(0:T)}\right\|^2 - \sigma h\left(\mathbf{G}^{(0)}\right)\right). \tag{37}$$

The first term is a sparsity prior and $h\left(\mathbf{G}^{(0)}\right)$ is the acyclicity constraint from Zheng et al. (2018).

The terms $\mathbb{E}_{q_\phi(\mathbf{Z}^{(1:T),n}|\mathbf{X}^{(1:T),n})}\left[-\log q_\phi\left(\mathbf{Z}^{(1:T),n}|\mathbf{X}^{(1:T),n}\right)\right], \mathbb{E}_{q_\phi(\mathbf{G})}[-\log q_\phi(\mathbf{G})]$ and $\mathbb{E}_{q_\phi(\mathbf{F})}[-\log q_\phi(\mathbf{F})]$ represent the entropies of the variational distributions and are evaluated in closed form, since their parameters are modeled as samples from Gaussian and Bernoulli distributions.

Finally, the prior term $p(\mathbf{F})$ is evaluated based on the assumed generative distribution mentioned in equation 3.

### B.3. Spatial Factors

As detailed in the main paper, we use spatial factors with RBF kernels to model the spatial variability in the data. To capture more complex spatial structures, we model the scale by introducing two additional parameter matrices $\mathbf{A}$ and $\mathbf{B}$. The matrix $\mathbf{A} = \begin{bmatrix} a & b \\ c & d \end{bmatrix}$ and the vector $\mathbf{B} = \begin{bmatrix} e \\ g \end{bmatrix}$ together influence the covariance structure of the RBF. Specifically, the covariance matrix $\Sigma$ is constructed as:

$$\Sigma = \mathbf{A}\mathbf{A}^T + \exp(\mathbf{B}), \tag{38}$$

This covariance structure enables the RBF to capture anisotropic scaling in different directions. The matrix $\mathbf{A}\mathbf{A}^T$ provides a base covariance matrix, while the exponential transformation of $\mathbf{B}$ ensures that the resulting matrix is positive definite. As a result, the RBF kernel, which determines the spatial factor $\mathbf{F}$, is defined as:

$$\mathbf{F}_{\ell d} = \exp\left(-\frac{1}{2}\|x_\ell - \boldsymbol{\rho}_d\|^2_{\Sigma^{-1}}\right), \tag{39}$$

where $x_\ell$ denotes the spatial coordinates of the $\ell^{\text{th}}$ grid point, and $\|x_\ell - \boldsymbol{\rho}_d\|^2_{\Sigma^{-1}} = (x_\ell - \boldsymbol{\rho}_d)^T \Sigma^{-1}(x_\ell - \boldsymbol{\rho}_d)$ represents a Mahalanobis distance, allowing the RBF to have a more sophisticated shape that depends on the learned covariance $\Sigma$.

#### B.3.1. CHOICE OF METRIC IN THE SPATIAL KERNELS

Spatiotemporal data can originate from various sources, take different forms, and represent a range of phenomena. As a result, the choice of distance metric depends on the application: for local phenomena, Euclidean distance may suffice, while global analyses may require metrics that account for the Earth's curvature. To accommodate this diversity, SPACY allows for flexible modeling by incorporating different metrics with the spatial kernels in the spatial factors.

In this work, we use the SPACY with the RBF kernel equipped with two different metrics based on the target dataset.

**Euclidean Distance.** For Cartesian coordinate systems underlying our synthetic datasets, we employ the standard Euclidean distance:

$$d_{\text{Euc}}(x_\ell, x_{\ell'}) = \|x_\ell - x_{\ell'}\|_2 = \sqrt{\sum_{k=1}^{K}(x_{\ell,k} - x_{\ell',k})^2} \tag{40}$$

where $K$ is the number of spatial dimensions.

**Haversine Distance.** For Earth-scale climate observations, we compute great-circle distances using the Haversine formula:

$$d_{\text{Hav}}(x_\ell, x_{\ell'}) = 2r \arcsin\left(\sqrt{\sin^2\left(\frac{\Delta\phi}{2}\right) + \cos\phi_i \cos\phi_j \sin^2\left(\frac{\Delta\lambda}{2}\right)}\right) \tag{41}$$

where $(\phi_i, \lambda_i)$ are latitude/longitude coordinates, $\Delta\phi = \phi_j - \phi_i$, $\Delta\lambda = \lambda_j - \lambda_i$, and $r$ is Earth's radius. This preserves the inherent spherical geometry of the climate dataset.

### B.4. Multivariate Extension

We extend SPACY to address multivariate observational time series, where $\mathbf{X} \in \mathbb{R}^{V \times L \times T}, V \geq 1$ represents variates or domains. We make the following changes to the model architecture. The observed timeseries $\left(\mathbf{X}_{1:L}^{(1:T)}\right)_{(1:V)} \in \mathbb{R}^{V \times L \times T}$ now includes observations from multiple variates, with $\mathbf{X}_{(v)}$ representing observed data specific to variate $v$.

**Forward Model.** For each variate $v$, we learn $D_v$ nodes, such that $D = \sum_{v=1}^{V} D_v$, where the $D_v$, $v \in V$ are input as hyperparameters. We learn separate spatial factors $\mathbf{F}_{(v)} \in \mathbb{R}^{L \times D_v}$ for each variate $v$. Let $\mathbf{Z}_{(v)} \in \mathbb{R}^{D_v \times T}$ denote the latent variables inferred from the $v^{\text{th}}$ variate. We perform causal discovery on the concatenated representations $\mathbf{Z} = \left[ \mathbf{Z}_{(1)}, \ldots, \mathbf{Z}_{(V)} \right] \in \mathbb{R}^{D \times T}$. In order to map from the latent representations to the multivariate observational space, we perform the tensor multiplication, defined below:

$$\mathbf{X}_{\ell,(v)}^{(t)} = g_\ell^{(v)} \left( \left[ \mathbf{F}_{(v)} \mathbf{Z}_{(v)} \right]_\ell^{(t)} \right) + \varepsilon_{\ell,(v)}^{(t)}, \quad \varepsilon_{\ell,(v)}^{(t)} \sim \mathcal{N}(0, \sigma_\ell^2 I) \tag{42}$$

where $g_\ell^{(v)}$ is the variate-specific nonlinearity. We implement $g_\ell^{(v)}$ as:

$$g_\ell^{(v)}(x) = \Xi \left( [x, \mathscr{E}_{\ell,(v)}] \right)$$

with learned embedding $\mathscr{E} \in \mathbb{R}^{V \times L \times f}$, where $f$ is the embedding dimension.

**Variational Inference.** To model the variational distribution of the latent variables, we use a separate multilayer perceptron (MLP) for each variate. Specifically, $\zeta_\mu^{(v)}$ computes the mean, and $\zeta_{\sigma^2}^{(v)}$ computes the log-variance of the variational distribution for the $v^{\text{th}}$ variate.

$$q_\phi \left( \mathbf{Z}_{(v)}^{(1:T)} | \mathbf{X}_{(v)}^{(1:T)} \right) = \mathcal{N} \left( \zeta_\mu^{(v)}(\mathbf{X}_{(v)}^{(t)}), \exp \left( \zeta_{\sigma^2}^{(v)}(\mathbf{X}_{(v)}^{(t)}) \right) \right).$$

## B.5. Training Details

We use an 80/20 training and validation split to evaluate the validation likelihood during training. We use an augmented Lagrangian training procedure to enforce the acyclicity constraint in the model (Zheng et al., 2018). We closely follow the procedure employed by Geffner et al. (2022); Gong et al. (2023) for scheduling the learning rates (LRs) across different modules of our model.

**Freezing Latent Causal Modules.** To stabilize the training and ensure accurate causal discovery, we freeze the parameters of the latent SCM and causal graph, and only train the spatial factors and encoder for the first 200 epochs. This allows the spatial factor parameters to be learned without interference from incorrect causal relationships in the latent space. Once these modules are unfrozen after 200 epochs, the complete forward model and variational distribution parameters are trained jointly for the rest of the training process.

## B.6. Evaluation Details

We use the mean correlation coefficient (MCC) as a measure of alignment between the inferred and true latent variables, widely used in causal representation learning works (Yao et al., 2022b;a). Here, MCC is computed as the mean of the correlation coefficients between each pair of true and inferred latent variables to measure how well the inferred latent time series match the true underlying latent time series.

To evaluate the accuracy of inferred causal graphs and representations, we find a permutation to match the nodes of the inferred graph to the ground truth. Specifically, we apply the Hungarian algorithm to find the optimal permutation of nodes that aligns the inferred graph's adjacency matrix with the ground truth, minimizing the discrepancies between them. This optimal permutation is then used to calculate both the F1 Score and the Mean Correlation Coefficient (MCC).

# C. Experimental Details

## C.1. Synthetic Datasets

This section provides more details about how we set up and run experiments using SPACY on synthetic datasets.

### C.1.1. DATASET GENERATION

The spatial decoder, represented by the function $g_\ell$, is configured to be linear or nonlinear, depending on the experimental setting. For the linear setting, $g_\ell$ is set to the identity function, while for nonlinear scenarios, we use randomly initialized MLPs. We generate $N = 100$ samples of data, with $T = 100$ time length each and represented on a grid of size $100 \times 100$. We vary the number of nodes ($D = 10, 20$ and $30$) in each setting.

For the ground-truth latent, we generate two separate sets of synthetic datasets: a linear dataset with independent Gaussian noise and a nonlinear dataset with history-dependent noise modeled using conditional splines (Durkan et al., 2019). We generate a random graph (specifically, Erdős-Rényi (ER) graphs) and treat it as the ground-truth causal graph. Specifically, we sample (directed) temporal adjacency matrices $\mathbf{G} \in \mathbb{R}^{(\tau+1) \times D \times D}$ with $4 \times D$ edges in the instantaneous adjacency matrix and $2 \times D$ connections in the lagged adjacency matrices. We regenerate the adjacency matrix until the instantaneous graph is a DAG.

**Linear SCM.** We model the data as:

$$\mathbf{Z}_d^{(t)} = \sum_{k=0}^{\tau} \sum_{d'=1}^{D} (\mathbf{G} \circ W)_{d',d}^k \times \mathbf{Z}_{t-k}^d + \eta_d^t \tag{43}$$

with $\eta_d^t \in \mathcal{N}(0, 0.5)$. Each entry of the matrix $W$ is drawn from $U[0.1, 0.5] \cup U[-0.5, -0.1]$

**Non-linear SCM** We model the data as:

$$\mathbf{Z}_d^{(t)} = f_d \left( \mathrm{Pa}_G^d(<t), \mathrm{Pa}_G^d(t) \right) + \eta_d^{(t)}$$

where $f_d$ are randomly initialized multi-layer perceptions (MLPs), and the random noise $\eta_d^{(t)}$ is generated using history-conditioned quadratic spline flow functions (Durkan et al., 2019). The MLPs for the functional relationships are fully-connected with two hidden layers, 64 units and ReLU activation. The history dependency is modelled as a product of a scale variable obtained by the transformation of the averaged lagged parental values through a random-sampled quadratic spline, and Gaussian noise variable. Each sample of the synthetic datasets is generated with a series length of 200 steps with a burn-in period of 100 steps.

**Spatial Factors** To generate the spatial factor matrices $\mathbf{F}$, we first sample the centers $\boldsymbol{\rho}_d$ of the RBF kernels uniformly over the grid while enforcing a minimum distance constraint to ensure separation between centers. Specifically, the minimum distance between any two centers is set to be $\frac{1}{10}$ of the grid dimension. The scales $\boldsymbol{\gamma}_d$ are sampled to define the extent of each RBF kernel, drawn uniformly from the range $U[3, 6]$. With these parameters, each entry of the spatial factor matrix $\mathbf{F}_d^\ell$ is determined by the RBF kernel as follows:

$$\mathbf{F}_{\ell d} = \exp \left( -\frac{||x_\ell - \boldsymbol{\rho}_d||^2}{\exp(\boldsymbol{\gamma}_d)} \right),$$

where $x_\ell$ denotes the spatial coordinates of the $\ell^{\text{th}}$ grid point, $\boldsymbol{\rho}_d$ is the center, and $\boldsymbol{\gamma}_d$ is the scale of the $d^{\text{th}}$ latent variable.

**Spatial Mapping** For the generation of $\mathbf{X}_\ell$, we pass the product of the spatial factors and the latent time series through a non-linearity $g_\ell$:

$$\mathbf{X}_\ell = g_\ell \left( [\mathbf{FZ}]_\ell \right) + \varepsilon_\ell, \quad \varepsilon_\ell \sim \mathcal{N}(0, \sigma_\ell^2 I) \tag{44}$$

where $g_\ell$ is the spatial mapping. It is implemented as a randomly initialized multi-layer perception (MLP) with the embedding of dimension 1 in the non-linear map setting, or as an identity function in the linear map setting. $\varepsilon_\ell$ is the grid-wise Gaussian noise added.

**Multivariate** For the multi-variate experiments, we consider a setting with $V = 2$ variates to evaluate the performance of SPACY in capturing inter-variate interactions. The latent dimension is set to $D = 10$, with each variate-specific spatial factor contributing independently to the observed data. We generate the datasets with $D_1 = D_2 = 5$ nodes, according to the forward model described in Appendix B.4.

### C.2. Baselines

For all baselines, we used the default hyperparameter values. We use the Mapped-PCMCI implementation from (Tibau et al., 2022)[2]. For Linear-Response we refer to the implementation from (Falasca et al., 2024)[3]. For LEAP and TDRL, we implemented the encoder and decoder using convolution neural networks as this choice best fits our data modality. For LEAP we used the CNN encoder and decoder architecture from the mass-spring system experiment. [4]. For TDRL we used

---

[2]Mapped-PCMCI: https://github.com/xtibau/savar
[3]Linear-Response: https://github.com/FabriFalasca/Linear-Response-and-Causal-Inference
[4]LEAP: https://github.com/weirayao/leap

the CNN encoder and decoder architecture from the modified cartpole environment experiment [5]. For TensorPCA-PCMCI we implement a model based on the formulation from (Babii et al., 2023) and PCMCI$^+$ (Runge, 2020b).

For multi-variate experiments (Synthetic, Climate) with Mapped-PCMCI, we run Varimax-PCA on each of the variates separately to obtain $D_v$ principal components for each $v \in [V]$. We concatenate the principal components to produce $D = \sum_{v=1}^{V} D_v$ nodes and perform causal discovery with PCMCI$^+$.

### C.3. Hyperparameters

In this section, we list the hyperparameters choices for SPACY in our experiments.

| Dataset | Synthetic ($D = 10, 20, 30$) | Synthetic-Multivariate | Global Temperature |
|---|---|---|---|
| **Hyperparameter** | | | |
| Matrix LR | $10^{-3}$ | $10^{-3}$ | $10^{-3}$ |
| SCM LR | $10^{-3}$ | $10^{-3}$ | $10^{-3}$ |
| Spatial Encoder LR | $10^{-3}$ | $10^{-3}$ | $10^{-3}$ |
| Spatial Factor LR | $10^{-2}$ | $10^{-2}$ | $10^{-2}$ |
| Spatial Decoder LR | $10^{-3}$ | $10^{-3}$ | $10^{-3}$ |
| Batch Size | 100 | 100 | 500 |
| # Outer auglag steps | 60 | 60 | 60 |
| # Max inner auglag steps | 6000 | 6000 | 6000 |
| $\xi_f, \lambda_f$ embedding dim | 64 | 64 | 64 |
| Sparsity factor $\alpha$ | 10 | 10 | 40 |
| Spline type | Quadratic | Quadratic | Quadratic |
| $g_\ell$ embedding dim | 32 | 32 | 32 |

Table 2: Table showing the hyperparameters used with SPACY.

For the Synthetic, Synthetic-Multivariate, and Global Climate datasets, the outer augmented Lagrangian (auglag) steps are set to 60, with a maximum of 6000 inner auglag steps. For Synthetic datasets we used batch size of 100 samples per training, and 500 for Global Climate datasets.

We used the rational spline flow model described in (Durkan et al., 2019). We use the quadratic rational spline flow model in all our experiments, with 8 bins. The MLPs $\xi_f$ and $\lambda_f$ have 2 hidden layers each and LeakyReLU activation functions. We also use layer normalization and skip connections. Table 2 summarizes the hyperparameters used for training.

### C.4. Visualization details

We visualize the spatial factors for both the synthetic and global temperature experiments by identifying regions where individual nodes exert significant spatial influence. For each node $d$, we threshold the spatial factors $\mathbf{F}_d$ by selecting values above a specified percentile (e.g., 95%), resulting in a binary mask that highlights areas of dominant activity. These masks are then aggregated to produce a comprehensive visualization of the spatial influence patterns across all nodes, revealing both their individual spatial footprints and areas of overlap.

For spatial factors with complex structure, we further simplify the representation by merging nearby nodes based on the proximity of their centers. Specifically, we compute pairwise distances between node centers and merge nodes whose distances fall below a percentile-based threshold (e.g., the lowest 5%), yielding a more interpretable depiction of the global spatial dynamics.

## D. Additional Results

### D.1. Qualitative Results

Figure 8 shows a comparison between the ground truth, inferred spatial factors by SPACY, spatial matrix $W$ inferred by CDSD, and mode weights by Mapped PCMCI on synthetic datasets. Overall, we observe that the inferred spatial factors

---
[5]TDRL: https://github.com/weirayao/tdrl

from SPACY align well with the ground truth nodes in terms of location and scales with minor deviations in shape. In the non-linear SCM dataset, the model shows some slight errors with at most 1 missing mode (for Non-linear SCM, Nonlinear Mapping). This is also reflected by the quantitative results as performance falls slightly for this setting.

The spatial matrix $W$ inferred by CDSD demonstrates overall strong performance, achieving competitive accuracy in linear settings. However, failures begin to emerge in non-linear configurations, where mode collapse occurs. This phenomenon leads to missing nodes and causes the spatial representations to scatter across the grid rather than localizing distinct modes.

The spatial factors, or mode weights as referred to in Tibau et al. (2022), recovered by Mapped PCMCI, perform competitively in linear mapping settings. However, similar to CDSD, its accuracy deteriorates when non-linear spatial transformations are applied, leading to artifacts and errors in node identification. In particular, for the Non-linear SCM and Nonlinear Mapping cases, Mapped PCMCI misidentifies several latent representations, with some modes being incorrectly placed inside others, highlighting its limitations in handling complex spatial structures.

### D.2. Multivariate Synthetic Dataset

| Setting | Method | F1 | MCC |
|---|---|---|---|
| Linear SCM, Linear $g_\ell$ | M-PCMCI | 0.334 | **0.969** |
| | T-PCMCI | 0.127 | 0.670 |
| | **SPACY** | $\mathbf{0.656}_{\pm 0.0}$ | $0.920_{\pm 0.1}$ |
| Linear SCM, Non-linear $g_\ell$ | M-PCMCI | 0.387 | **0.924** |
| | T-PCMCI | 0.115 | 0.660 |
| | **SPACY** | $\mathbf{0.596}_{\pm 0.1}$ | $0.834_{\pm 0.0}$ |
| Non-linear SCM, Linear $g_\ell$ | M-PCMCI | 0.178 | 0.883 |
| | T-PCMCI | 0.032 | 0.644 |
| | **SPACY** | $\mathbf{0.461}_{\pm 0.0}$ | $\mathbf{0.895}_{\pm 0.1}$ |
| Non-linear SCM, Non-linear $g_\ell$ | M-PCMCI | 0.170 | 0.807 |
| | T-PCMCI | 0.039 | 0.573 |
| | **SPACY** | $\mathbf{0.509}_{\pm 0.1}$ | $\mathbf{0.825}_{\pm 0.1}$ |

Table 3: Results on different configurations of the multi-variate synthetic datasets. M-PCMCI and T-PCMCI are the respective baselines Mapped-PCMCI (Tibau et al., 2022) and TensorPCA-PCMCI (Babii et al., 2023). We report the F1 and MCC scores for latent dimension $D = 10$. Average over 5 runs reported

For multi-variate settings, we include the two-step baseline TensorPCA + PCMCI$^+$ (Babii et al., 2023) in addition to Mapped PCMCI. The results of multi-variate synthetic experiments are shown in Table 3. SPACY maintains its performance advantage in the multi-variate setting, where it consistently outperforms the baselines in terms of causal discovery F1 score. Similar to the univariate case, Mapped-PCMCI performs well in terms of MCC but fails to recover the causal links accurately. SPACY outperforms Mapped-PCMCI in terms of both MCC and F1 score in the non-linear SCM settings. TPCA-PCMCI falls short in all settings. Figure 9 provides a visual illustration of the recovered spatial factor for multi-variate setting for SPACY.

### D.3. Ablation Study

**Over-specifying $D$.** SPACY requires specifying the number of latent variables $D$ as a hyperparameter. In practice, the exact number of underlying factors is often unknown. We examine the effect of overspecifying $D$ by setting it to $D^* + 10$, where $D^*$ represents the true number of nodes used to generate the data. Additionally, we explored the robustness of the model over different levels of parameterizations with $D^* + n$, $n \in \{0, 2, 5, 7, 10\}$ We use the synthetic dataset with grid dimensions $100 \times 100$, linear SCM and non-linear mapping.

Figure 10 illustrates the results of our experiment. When $D^* = 10$, despite over-specifying the number of nodes, the inferred spatial modes' general locations align well with the ground truth. The presence of additional modes does not significantly detract from the accuracy of detecting the primary spatial modes. This suggests that SPACY maintains robust learning of latent representations even when $D$ exceeds the true number of spatial factors. This observation also holds true when

comparing the causal discovery performance using the F1 score.

Figure 11 illustrates the results of our exploration on different levels of over-parameterizations. As the latent dimension increases, the performance does not degrade. This suggests that SPACY maintains robust learning of latent representations across different levels of over-parameterization, offering flexibility in the choice of the hyperparameter $D$ when there is uncertainty about the true latent dimensionality.

**Different Kernels**    We experiment with different spatial kernel functions in order to test SPACY's robustness to the choice of kernel used in modeling the spatial factors. We use the synthetic dataset with linear SCM and nonlinear spatial mapping, where the ground truth spatial mapping is generated using RBF kernels.

The Matérn kernel is a generalization of the Radial Basis Function (RBF) kernel and is widely used in spatial statistics and machine learning due to its flexibility in modeling functions of varying smoothness. The Matérn kernel is defined as:

$$k_{\text{Matérn}}(r) = \frac{2^{1-\nu}}{\Gamma(\nu)} \left( \sqrt{2\nu} \frac{r}{s} \right)^{\nu} K_{\nu} \left( \sqrt{2\nu} \frac{r}{s} \right),$$

where:

- $r = \|\boldsymbol{\ell} - \boldsymbol{\ell}'\|$ is the Euclidean distance between points $\boldsymbol{\ell}$ and $\boldsymbol{\ell}'$,

- $s$ is the length scale,

- $\nu > 0$ controls the smoothness of the function,

- $\Gamma(\cdot)$ is the gamma function,

- $K_{\nu}(\cdot)$ is the modified Bessel function of the second kind.

For specific values of $\nu$, the Matérn kernel simplifies to closed-form expressions:

- **When $\nu = 1.5$:**

$$k_{\text{Matérn}}^{1.5}(r) = \left( 1 + \frac{\sqrt{3}r}{s} \right) \exp \left( -\frac{\sqrt{3}r}{s} \right).$$

- **When $\nu = 2.5$:**

$$k_{\text{Matérn}}^{2.5}(r) = \left( 1 + \frac{\sqrt{5}r}{s} + \frac{5r^2}{3s^2} \right) \exp \left( -\frac{\sqrt{5}r}{s} \right).$$

These formulations allow us to model functions with different degrees of smoothness, providing a more flexible approach compared to the RBF kernel.

We test SPACY with the Matérn kernel using two settings: $\nu = 1.5$ and $\nu = 2.5$. Figure 12 presents the F1-Score and MCC for SPACY using the RBF kernel and both Matérn kernel settings. The results show that SPACY achieves similar performance with the Matérn kernels compared to the RBF kernel, indicating that the variational inference framework effectively generalizes across different kernel functions. The inferred spatial modes' general locations and scales align well with the ground truth across all kernel settings (illustrated in Figure 13). This consistency demonstrates that SPACY's spatial representations are robust to the choice of kernel function.

**How well do the RBF Kernels approximate anisotropy?**    While SPACY's theoretical framework supports fully anisotropic spatial kernels, we use RBF kernels in our experiments since they parsimoniously capture high-level features. In this experiment, we verify SPACY's robustness when modeling data generated from non-isotropic spatial factors using only isotropic RBF kernels. To create challenging test conditions, we first generate synthetic data with irregular anisotropic spatial factors defined as:

$$F_i((x,y)) = \exp \left( -\frac{(u(x,y) - x_0^{(i)})^2}{2(\sigma_x^{(i)})^2} - \frac{(v(x,y) - y_0^{(i)})^2}{2(\sigma_y^{(i)})^2} \right) \tag{45}$$

where $x_0$ and $y_0$ denote fixed points corresponding to the 'center' of the factors, and the coordinates are warped via sinusoidal transformations:

$$u(x, y) = x + A^{(i)} \sin(\omega^{(i)}(x - x_0^{(i)})) \sin(\omega^{(i)}(y - y_0^{(i)}))$$
$$v(x, y) = y + A^{(i)} \cos(\omega^{(i)}(x - x_0^{(i)})) \cos(\omega^{(i)}(y - y_0^{(i)}))$$

Here, $A^{(i)}$ and $\omega^{(i)}$ control the amplitude and frequency of spatial distortion for the $i$-th latent factor. We test with sampled warp parameters ($A \in \text{Uniform}([2, 4]), \omega \in \text{Uniform}([0.1, 0.3])$) to simulate complex neural response fields. The results are presented in Figure 14, which demonstrates that SPACY achieves accurate localization and scale estimation for latent variables, even under irregular and anisotropic spatial factors. The high F1-Score and MCC further validate SPACY's robust performance when using RBF-based approximation on irregularly structured data, confirming its ability to reconstruct the underlying causal graph effectively.

### D.4. Global Climate

The **Global Climate Dataset** is a comprehensive, mixed real-simulated dataset encompassing monthly global temperature and precipitation data spanning the years 1999 to 2001. It contains 7531 simulated samples, each over a time period of 24 months, covering the entire globe at a fine spatial resolution. The grid size is $145 \times 192$, which corresponds to a spatial division of approximately $1.24°$ latitude and $1.875°$ longitude. This spatial resolution allows the dataset to provide detailed global coverage, capturing temperature and precipitation variations across diverse geographical regions. The resulting data dimensions are $7531 \times 2 \times 24 \times 145 \times 192$, representing the total number of samples, variates (temperature and precipitation), the temporal sequence, and the spatial grid, respectively.

To facilitate causal analysis of complex climate phenomena beyond seasonal patterns, we apply a de-seasonality procedure. This normalization process involves computing the monthly mean for each month across all years and then subtracting the mean from the data of the corresponding month (for example, normalizing all January data by the mean of all January values).

For our analysis, we use the nonlinear variant of the SPACY method to uncover latent representations within the data. Specifically, we use 50 latent variables (denoted as $D = 50$) and a maximum lag of six months ($\tau = 6$). We set $D_1 = D_2 = 25$ latent variables. Figure 15 shows the inferred spatial modes across the different nodes, and the causal graph. We observe that the inferred modes are spatially confined, each with a distinct center and scale, which leads to improved interpretability.

Figure 16 presents the visualization of the modes and causal graph inferred by Mapped-PCMCI. The recovered spatial factors capture spatial correlations within each variate, revealing diffuse locality patterns in some regions such as Australia, Africa, and East Asia. This pattern is most noticeable in the temperature variate, where localized structures emerge. However, most inferred modes appear dispersed across the map, especially in the precipitation variate, suggesting that the underlying spatial structure lacks clear partitioning into distinct, interpretable modes.

Figures 17 and 18 show the node-wise visualization of modes inferred by Mapped-PCMCI, with demonstrations of individual inferred modes for both variates. The visualizations show that many modes recovered by Varimax are diffuse, uninterpretable, and lack clear physical locations, with clusters (e.g., node 42, 44) suggesting similar underlying components.

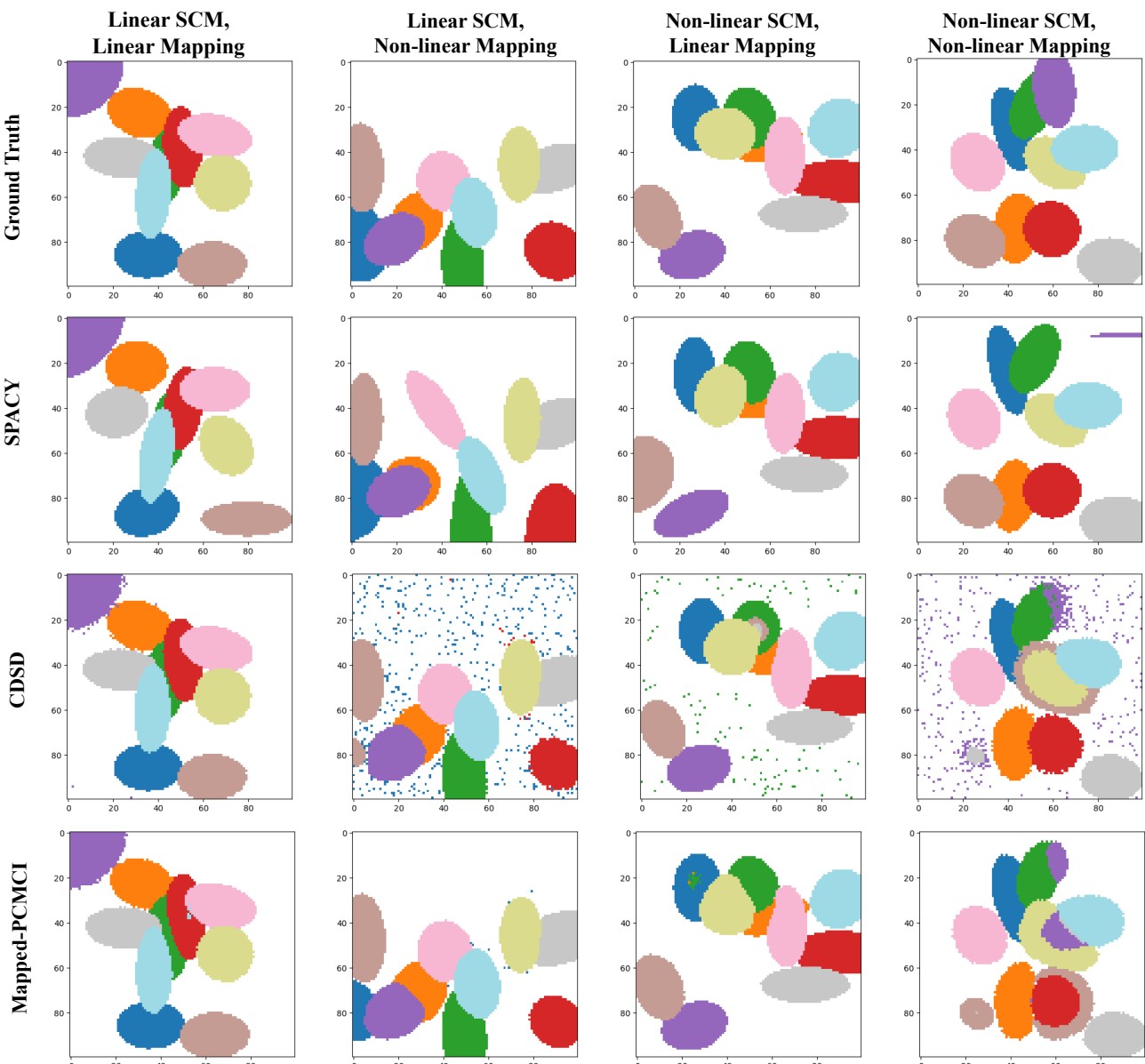

Figure 8: Visualization of the ground-truth and inferred spatial factors for different combinations of linear and nonlinear functions for SCMs and spatial mappings (top row: ground-truth, second row: inferred by SPACY, third row: inferred by CDSD (Boussard et al., 2023; Brouillard et al., 2024), bottom row: computed by Mapped-PCMCI (Tibau et al., 2022)). We demonstrate the visualization when latent dimension $D = 10$

.

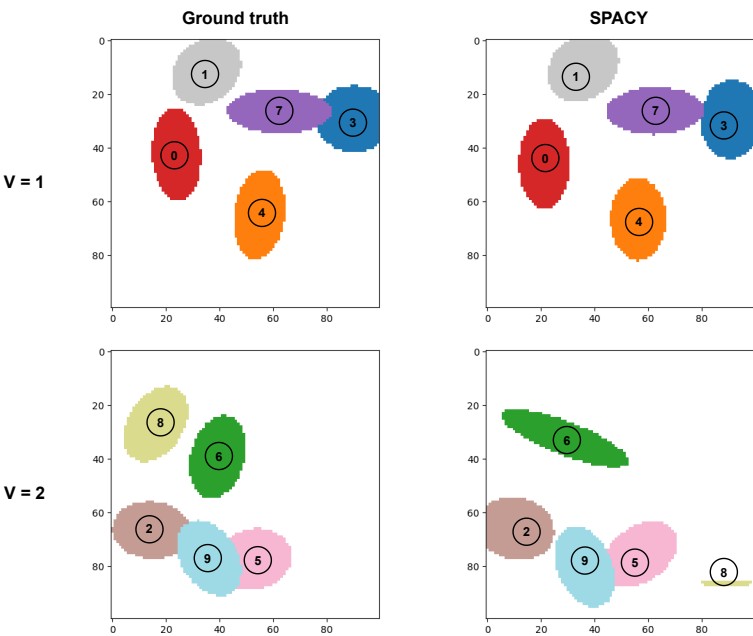

Figure 9: Example visualization of the ground-truth and inferred spatial factors for multi-variate synthetic dataset(top row: first variate, bottom row: second variate). We demonstrate the visualization where latent dimension D = 10, Linear SCM and Non-linear Mapping

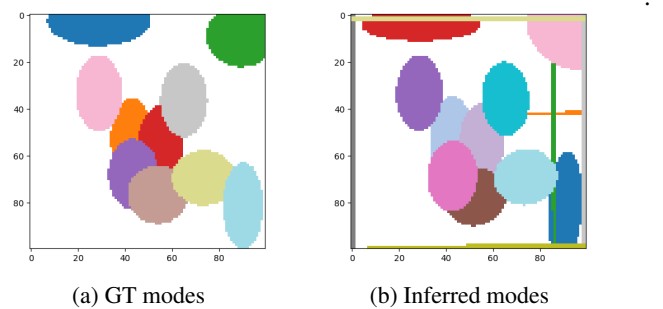

(a) GT modes

(b) Inferred modes

| $D^*$ | F1 score $(D = D^*)$ | F1 score $(D = D^* + 10)$ |
|---|---|---|
| 10 | $0.623 \pm 0.06$ | $\mathbf{0.642 \pm 0.07}$ |
| 20 | $\mathbf{0.752 \pm 0.03}$ | $0.549 \pm 0.03$ |
| 30 | $\mathbf{0.596 \pm 0.05}$ | $0.529 \pm 0.06$ |

(c) Causal discovery performance of SPACY-L

Figure 10: Overview of the results for over-specification ablation study. (a) Visualization of the ground-truth location and scale of the spatial modes. (b) Visualization of the inferred location and scale when we over-specify the number of nodes. (c) Causal discovery performance after matching and eliminating nodes. Average over 5 seeds reported

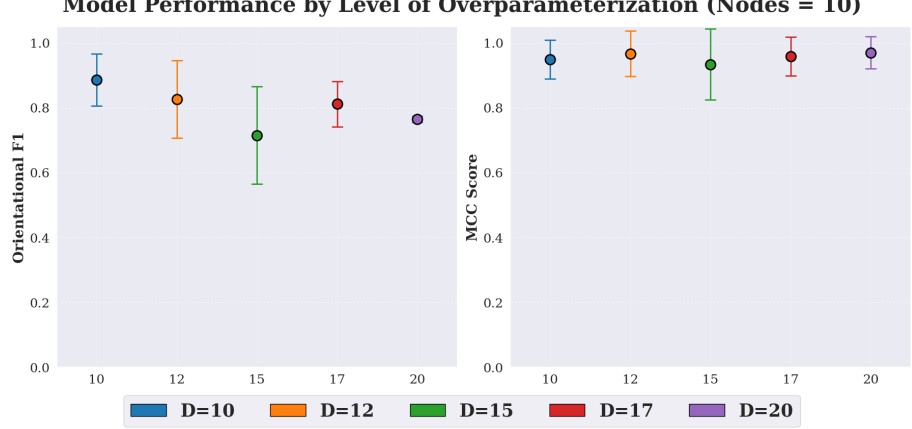

Figure 11: Different levels of over-specification ablation study. Average of 5 seeds reported

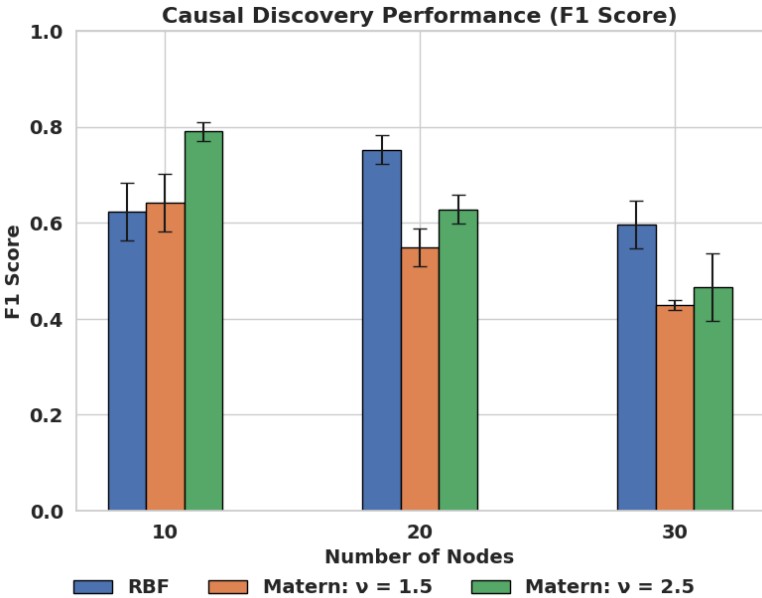

Figure 12: The causal discovery performance (F1 score) of SPACY using different kernel functions as spatial factors. Average of 5 seeds reported

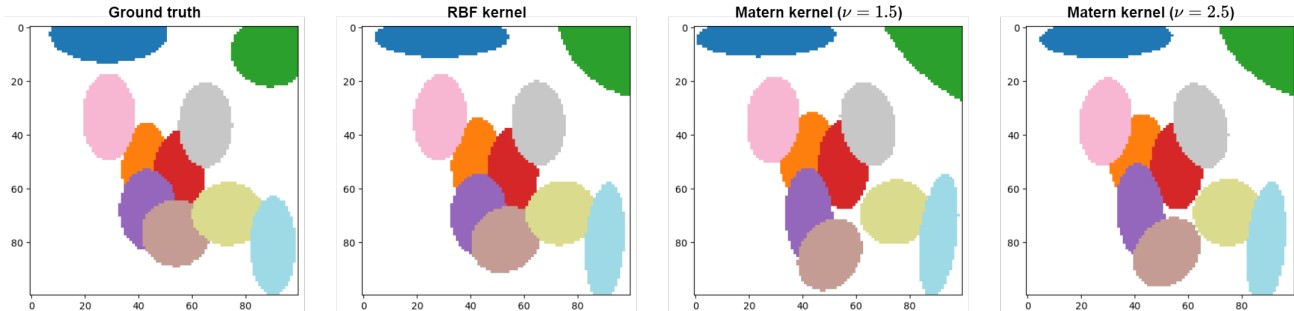

Figure 13: Overview of the visualization of the spatial factor when using different kernel functions. We compare inferred spatial factors using RBF, Matern Kernel ($\nu = 1.5$), and Matern Kernel ($\nu = 2.5$) with the ground truth spatial factors

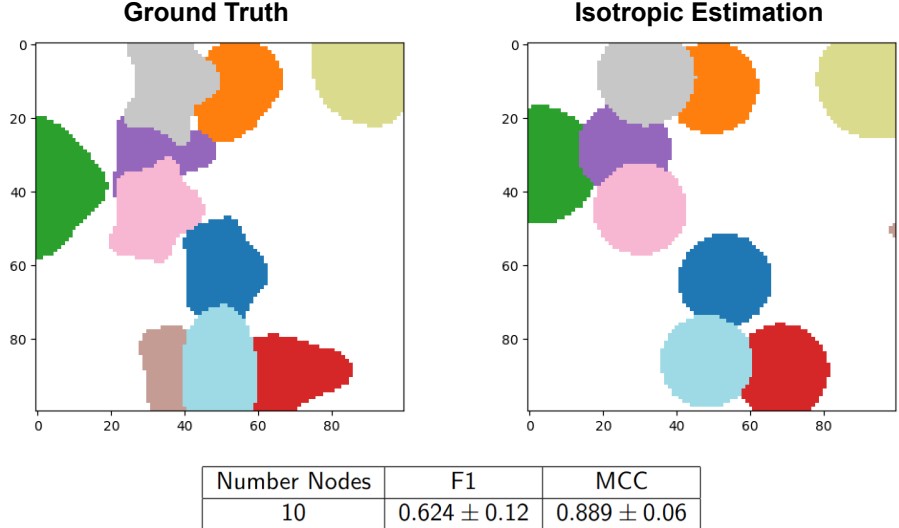

| Number Nodes | F1 | MCC |
|---|---|---|
| 10 | $0.624 \pm 0.12$ | $0.889 \pm 0.06$ |

Table: SPACY isotropic esimation on data with irregular spatial factors, data setting: MLP SCM + MLP spatial map

Figure 14: Visualization and results of the ground-truth spatial factors with irregular kernels, and the isotropic estimation by SPACY. Average of 5 seeds reported

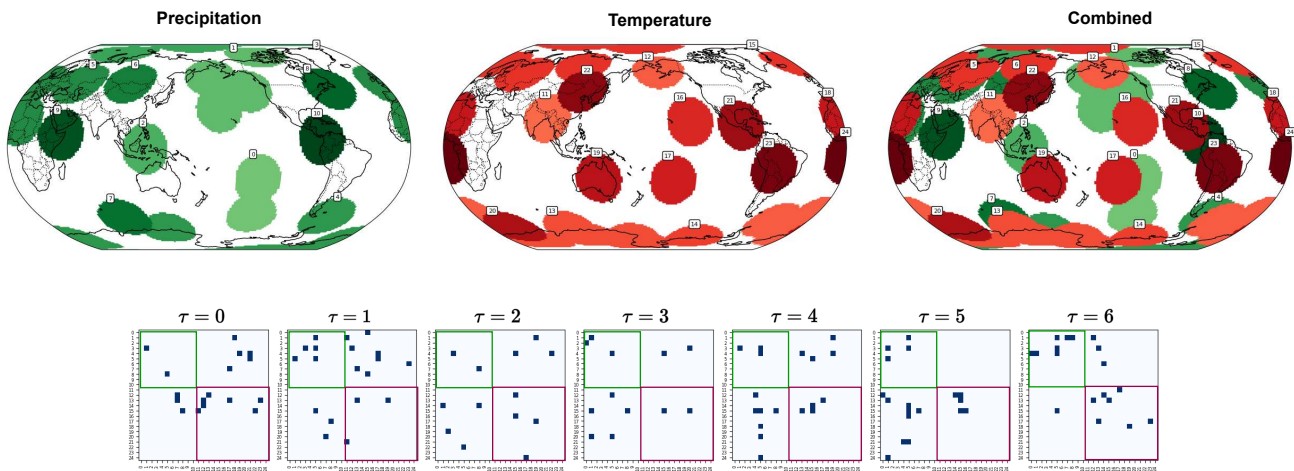

Figure 15: Visualization of the learned spatial factors and causal graph from Global Climate Dataset, after merging based on proximity and graph links. The spatial factor is demonstrated across precipitation (left), temperature (middle), and combined (right).

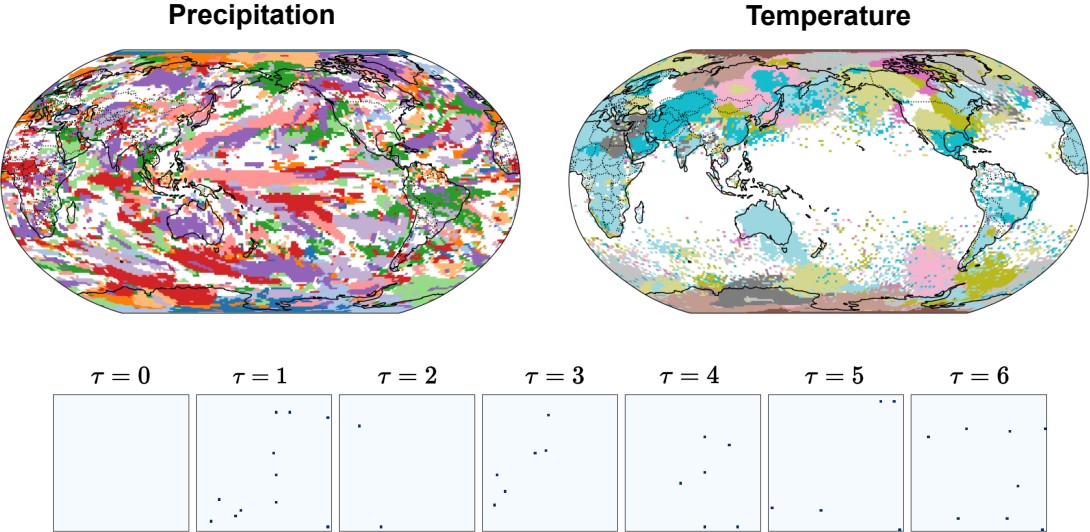

Figure 16: Visualization of the spatial factor inferred by Varimax-PCA and causal graph inferred by PCMCI+ from the climate dataset, following the procedure in Appendix C.4

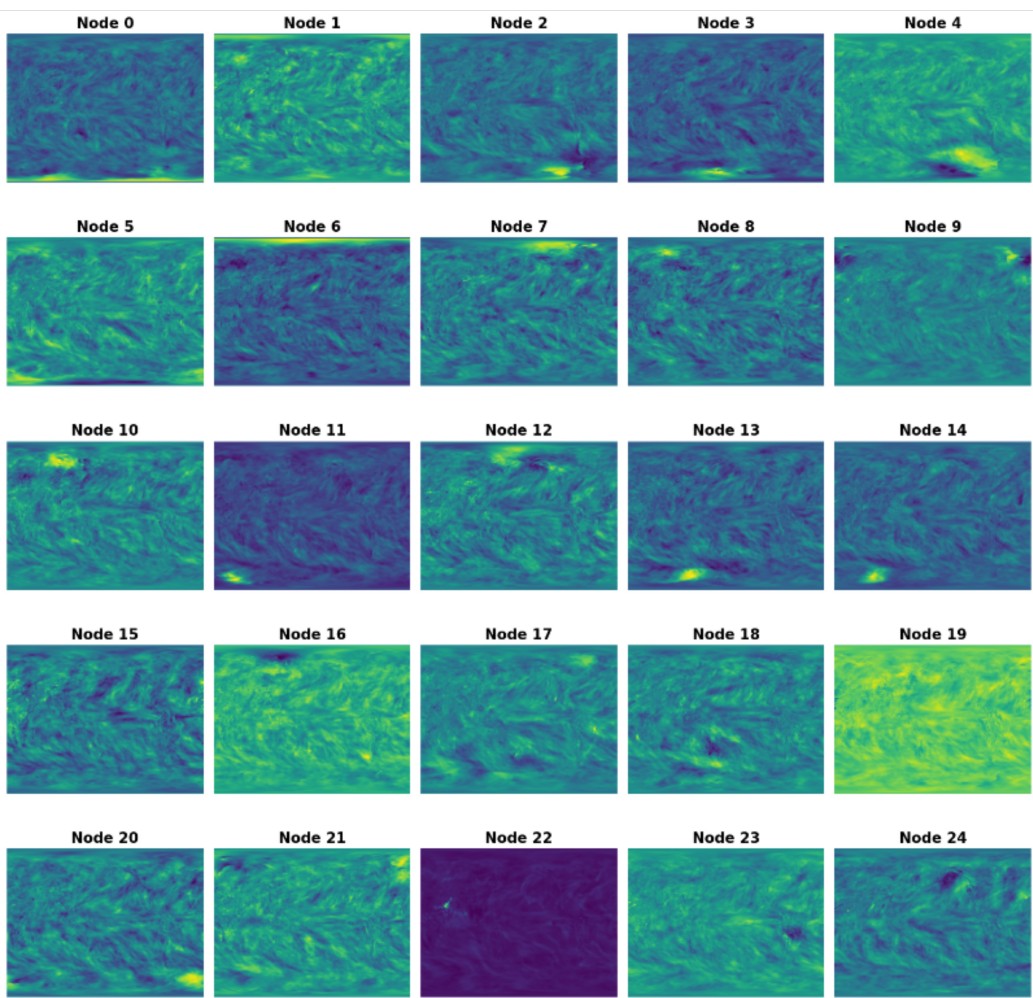

Figure 17: Visualization of the individual spatial nodes inferred by Varimax-PCA from the Climate Dataset. Precipitation variate displayed

# Varimax Visualization: Temperature

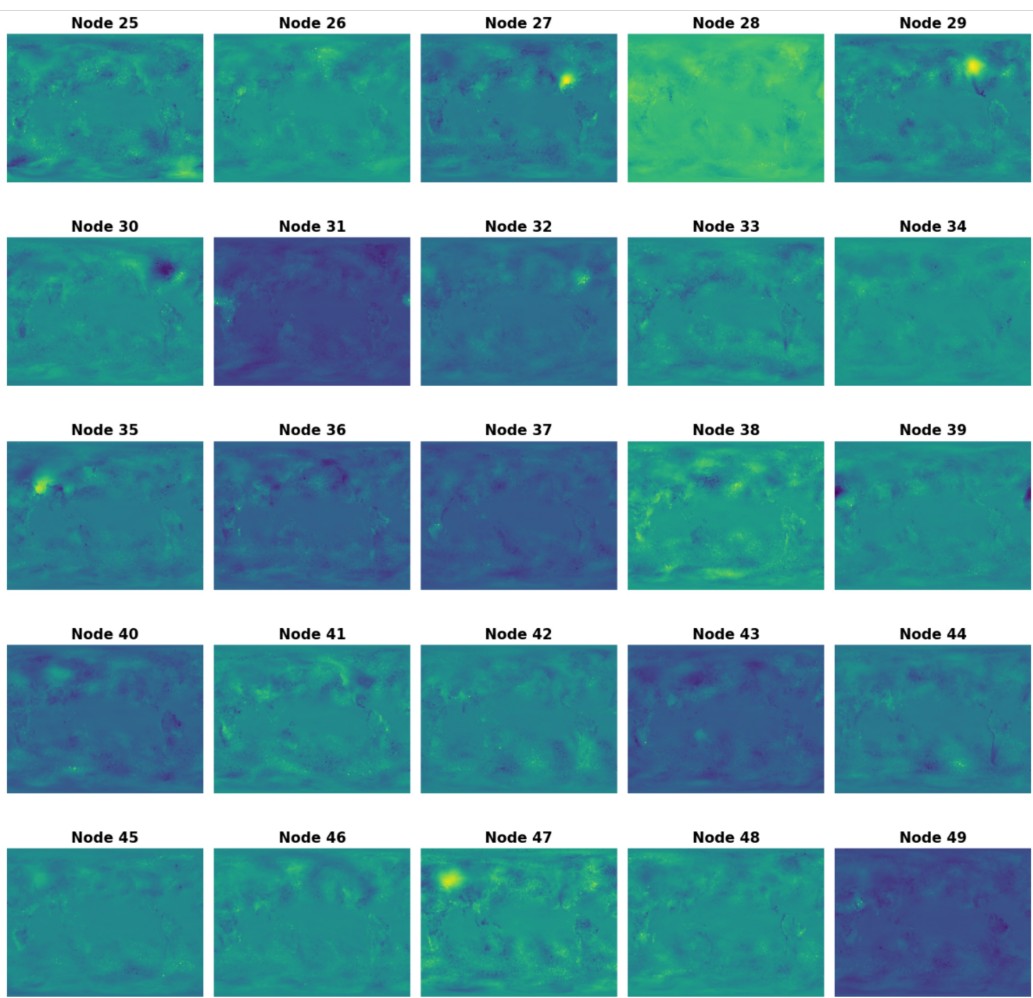

Figure 18: Visualization of the individual spatial nodes inferred by Varimax-PCA from the Climate Dataset. Temperature variate displayed

