# OpenReview forum: "Discovering Latent Causal Graphs from Spatiotemporal Data"
_ICML.cc/2025/Conference — ICML 2025 poster_

### Official Review · Reviewer_MCuj · 2025-03-12

**Overall Recommendation:** 2

**Summary:**

This manuscript analyzes spatiotemporal data with the goal of finding a temporal structural causal model of underlying latents.  This framework could be used for causal discovery of earth systems.  The manuscript includes proof of identifiability of reasonable assumptions, and provides empirical results on real and synthetic data.

**Claims And Evidence:**

I do not find the case study on the hybrid real-synthetic earth system overly convincing, as it is not clear that the inferred shapes with clear boundaries are realistic based on earth systems, nor was prediction really considered.

**Essential References Not Discussed:**

N/A

**Experimental Designs Or Analyses:**

As described above, I would like to see results on prediction.

An additional consideration is the impact of spatial factor shape on performance.  It is clear that SPACY outperforms competing approaches on the synthetic data, and is robust to mild permutations of the kernel in SPACY.  However, in practice I would expect more complex shapes that are non-isotropic, whereas the data and kernels are both isotropic, and I would like to see a clearer description of that result.

The claims on the climate datasets are too strong.  For instance, the relationships shown in Figure 6 are interesting in the fact that they recover known relationships, but the strong prior on the shape of spatial factors hinder interpretation.  In particular, the physics of the earth system requires that the relationship goes between the intermediate spatial locations, which is clearly missed here.  As such, it is difficult to assess whether these conclusions are good or oversimplified.  Part of this could be addressed by assessing forecasting strength to show that you are copying information.

Additionally, the claim that the SPACY factors are easier to interpret than the Varimax-PCA is difficult to evaluate, as the Varimax-PCA factors show known relationships that respect geographical changes in the earth.

**Methods And Evaluation Criteria:**

One thing that is clearly missing in the evaluation is how well these models predict.  An ideal model would both discover potential causal relationships but also explain a significant amount of variance in the system. In other words, if a system loses a lot of predictive ability, I would have a difficult time accepting its underlying latent model.  As such, I would encourage the authors to clearly describe how good their predictions are and compare this to the field at large.

**Other Comments Or Suggestions:**

Some other minor comments and suggestions:

SCM isn’t accounting for time in its initial definition of 3, which is confusing. It looks like you are defining these as independent processes in the preliminaries, which then moves to a temporal process in the Latent SCM.  This should be revised and clarified.

\ell is not defined in section 3.  I assume it’s the spatial index, but this should be explicitly stated.

G_j,d^k is not fully defined in equation 2.

The use of Rhino is not fully motivated or elucidated.

The definition of space used seems suboptimal.  While not just define space on a surface of a sphere?

The causal graph used in the method is not restricted to or encouraged to be a DAG, whereas an SCM is.  This choice should be discussed.

The generation of the synthetic data is not clear enough.  For instance, Erdos-Reyni is not typically directed, please describe given the SCM setup. Additionally, the generation through time should be clarified.

**Other Strengths And Weaknesses:**

Overall, I like that this is approaching a novel problem and attempting to make insights on complicated systems.

**Questions For Authors:**

Primarily, I want the authors to provide more details on robustness to non-isotropic shapes, as in real-world systems, and address how much variability the model captures of the real data.

**Relation To Broader Scientific Literature:**

The contributions seem appropriate given the contributions of the paper; albeit the evidence for the 3rd claim (empirical) is weaker than I would like.

**Theoretical Claims:**

The theoretical claims appear correct and the basis of the theory is well-founded in the literature.

---

> ### Author Rebuttal · Authors · 2025-04-01
>
> We thank the reviewer for a thorough and insightful review.
>
> **Predictive Performance Evaluation**: Our evaluation is consistent with the literature of causal discovery. As noted or observed in prior works [1, 2, 3] causal discovery/representation learning differs from prediction/forecasting. Causes may not explain a large portion of the effect’s variance. For example, let $X \sim N(0, 1)$ and $Y = 0.2X + E$ with $E \sim N(0, 1)$. X has a significant causal effect on Y yet explains only $\frac{0.2^2}{0.2^2 + 1} \approx 3.8 \\%$ of Y’s variance.
>
> Nevertheless, following the reviewer’s suggestion, we modify SPACY to predict one-step ahead $\widehat{X}^{t+1}$ using ${X}^{t}... {X}^{t-\tau}$. We edit the SCM module to autoregressively predict the latent values $\widehat{Z}^{t+1}$ from ${Z}^{t}... {Z}^{t-\tau}$ following the topological ordering of the inferred causal graph and then decode these to obtain the prediction. Similarly, we use the learned transition prior in LEAP to predict one step ahead. We report the results for our method on both the [synthetic](https://pasteboard.co/C0spUI2cWDJS.png) (Nonlinear SCM and Nonlinear spatial mapping) and the [climate](https://pasteboard.co/tH88MFprECMB.png) dataset.
>
> Our results indicate that SPACY captures most of the variance in the synthetic dataset. In the climate dataset, where complex exogenous factors dominate, the explained variance, though significant, is lower—consistent with expectations. LEAP shows a similar pattern, with lower performance than SPACY. In summary, while predictive performance is relevant in some contexts, our primary contribution is causal representation learning which we evaluate accordingly.
>
> **Non-Isotropic Spatial Factors**: Our experiments use isotropic and elliptical anisotropic (Section B.3) spatial kernels, but SPACY’s theoretical framework supports fully anisotropic kernels, as our identifiability guarantees require only mild conditions about the spatial factors. We used RBF kernels since they parsimoniously capture high-level features. However, our framework can also be used with more expressive kernels.
>
> To assess SPACY’s robustness to anisotropy, we conduct an additional ablation study in which the spatial factors are irregular and anisotropic. We introduce anisotropy by applying a sinusoidal warp to the spatial coordinates before using an anisotropic RBF kernel to generate data. We use isotropic RBF kernels with SPACY to model the synthetic data (Nonlinear SCM and Nonlinear spatial mapping setting). [Results](https://pasteboard.co/dKiKosZtDxdJ.png)
>
> Our findings show that even with isotropic RBF kernels, SPACY can approximate each latent variable’s location and scale and recover the causal graph. We will add these results to our paper.
>
> **Climate Dataset Claims**: We appreciate the reviewer’s concern regarding spatial factor interpretability. However, many key atmospheric processes (the Walker circulation, monsoonal systems, and ENSO teleconnections) exhibit strong localization due to the interplay between heterogeneous boundary conditions (e.g., land-sea contrast, topography) and localized forcings. Despite using isotropic kernels, our method captures these localized relationships, offering greater physical interpretability than traditional approaches (e.g., principal components or EOFs) that may obscure spatial meaning through averaging. Additionally, our framework avoids diffuse spatial factors, linking causal links to identifiable physical processes.
>
> **SPACY vs Varimax Interpretability**: SPACY’s spatial factors enforce geographic coherence by linking causal variables to interpretable regions, unlike Varimax’s scattered loadings. Visualizations show that many nodes recovered by Varimax are diffuse, uninterpretable, and lack clear physical locations, with clusters (e.g., nodes 42, 44) suggesting similar underlying components.
>
> [Precipitation](https://pasteboard.co/PV7yQmWl3Xh6.png), [Temperature](https://pasteboard.co/jVaQSkHfqFMS.png)
>
> **Other Suggestions:** We will clarify the notations and synthetic data generation process. To address some concerns:
>
> - Any differentiable temporal causal discovery algorithm can be used with SPACY. We chose Rhino due to its flexibility and strong identifiability guarantees.
> - SPACY operates with general metrics defined on $[0, 1]^K$. We can model diverse scenarios, such as the spherical grid with Haversine distance in the climate experiment.
> - Following [4], we use a DAGness constraint for the instantaneous matrix (Appendix C.1).
>
> **References**
>
> [1] Shmueli, Gt. "To explain or to predict?." (2010)
>
> [2] Lee, SY, et al. "Assessment of the predictive power of a causal variable: An application to the Head Start impact study." (2022)
>
> [3] Chauhan, R.S, et al. "Causation versus prediction in travel mode choice modeling." (2025)
>
> [4] Gong, W., et al. "Rhino: Deep Causal Temporal Relationship Learning with History-dependent Noise." (2022)

---

### Official Review · Reviewer_u93e · 2025-03-13

**Overall Recommendation:** 4

**Summary:**

The paper presents a causal discovery method called SPACY for structural causal models over spatiotemporal data embedded in grids. An identifiability theory is developed, and experiments compare to both structure-learning and causal representation learning models on both synthetic data and climate data.

**Claims And Evidence:**

Yes.

**Essential References Not Discussed:**

Handled in author response.

**Experimental Designs Or Analyses:**

I have checked the validity of these experimental designs, and they live up to the standard for the generative modeling field.  The authors have also added more experiments with additional datasets in the course of their response to reviewers.

**Methods And Evaluation Criteria:**

Yes.

**Other Comments Or Suggestions:**

I thank the authors for their engagement during the review and discussion process.

**Other Strengths And Weaknesses:**

Like several other papers submitted to ICML this year, the paper has the strength of giving novel identifiability theorems, which can help to make up for having only one non-synthetic experiment.

**Questions For Authors:**

N/A

**Relation To Broader Scientific Literature:**

The paper embeds itself well in the broader literature via a related works section.  As part of the author response, the authors have agreed to include the additional reference I recommended as part of their citations on RBF kernels.

**Theoretical Claims:**

The proofs are in the appendix and so I haven't checked them.

---

> ### Author Rebuttal · Authors · 2025-04-01
>
> We thank the reviewer for their careful review and for pointing us to the relevant prior work [1]. We will revise the manuscript to include this citation in the discussion of RBF kernels.
>
> We envision SPACY being applicable across various fields—for instance, in neuroscience for analyzing brain imaging data, or in epidemiology for identifying patterns in disease spread. We plan to include a discussion highlighting these potential applications, while leaving their in-depth exploration for future work.
>
> **References**
>
> [1] Sennesh, Eli, et al. "Neural topographic factor analysis for fmri data." Advances in Neural Information Processing Systems 33 (2020): 12046-12056.

---

### Official Review · Reviewer_WcLB · 2025-03-14

**Overall Recommendation:** 3

**Summary:**

This paper proposes a spatiotemporal causal discovery framework named SPACY based on the method of variational inference. SPACY introduces spatial kernel functions, aggregates spatially adjacent points by utilizing these kernel functions, maps the observed time series to latent representations, and discovers the causal structure in the low-dimensional latent space. Meanwhile, it proves the identifiability of the model in the continuous spatial domain, and verify the effectiveness of the proposed  method in synthetic data and real-world data.

**Claims And Evidence:**

The claims are well supported by the theoretical analysis and extensive experiments.

**Essential References Not Discussed:**

I am not sure if all key references are well-discussed.

**Experimental Designs Or Analyses:**

Experimental designs and analyses are sound.

**Methods And Evaluation Criteria:**

The proposed method is generally sound.

**Other Comments Or Suggestions:**

NAN

**Other Strengths And Weaknesses:**

Strengths:

1、 The paper is clearly written and well-organized.

2、 The paper proves the identifiability of the latent factors in the continuous spatial domain without relying on traditional assumptions, such as the assumption of sparsity of the causal graph.

3、 The proposed methods can explain the causal relations in the Global Climate Dataset, which shows a meaningful real-world application.



Weaknesses:

1. The assumption of Gaussian noise in the latent structural causal model (SCM) appears relatively restrictive, and the variational inference approach hinges on this condition. I am not sure if the proposed method is sensitive to this condition.


2. Theorems 1 and 2 establish the identifiability of the latent representations, yet the intuition behind identifying the causal structure among latent factors remains unclear. Could the authors provide a clearer explanation or illustrative example to bridge this gap?


3. The paper assesses partial robustness via hyperparameter experiments (e.g., overparameterization of D), but the question of how to systematically determine hyperparameter values to balance model complexity and performance remains unresolved. Further clarification or guidelines would strengthen this aspect.

**Questions For Authors:**

See Weakness.

**Relation To Broader Scientific Literature:**

The proposed method is important for the identification of causal representation in the time series data.

**Theoretical Claims:**

The theoretical claims are well-discussed.  But I am not family with the topic, I cannot ensure the proof is correct.

---

> ### Author Rebuttal · Authors · 2025-04-01
>
> We thank the reviewer for their careful review and thoughtful feedback.
>
> **Noise at latent vs. observation space**:  We clarify that there are **two separate additive noises** at latent and observation space respectively. The Gaussian noise assumption applies only to the **observation space**, not the latent SCM. The identifiability guarantees in Theorems 1–2 do not require explicit assumptions on the noise type in the latent SCM. Empirically, we validated our method on data with non-Gaussian and history-dependent noise (see experiments in Section 5). We will emphasize this distinction in the revised manuscript for better clarity.
>
> Furthermore, the Gaussian assumption in the observation space can be relaxed quite easily (see Step 1 in Appendix B.2.2 from [1]). We can also relax the assumption of Gaussian noise in Theorems 1 and 2 using the aforementioned technique in the updated manuscript.
>
> **Causal Structure Identification**: Theorems 1 and 2 focus on establishing the identifiability of the latent variables from observational data. **Once the latents are identified, any causal discovery algorithm with identifiability guarantees (such as Rhino, which we used in our paper) can recover the causal graph**, since the causal relationships are encoded in the conditional independence relationships of the latent variables. We will clarify this in the Theory section of our paper.
>
> **Choice of hyperparameters**: We thank the reviewer for highlighting this practical concern. Our experiments (Section 5.3) demonstrate that overparameterization the latent dimension D does not degrade performance, allowing users to err on the side of overestimation. To provide systematic guidelines for using our model, we will:
> 1. Add a recommendation in Section 5.3 to select D larger than the anticipated latent dimensionality.
> 2. [Include an additional experiment with a plot demonstrating the performance trend based on different levels of over-parameterization](https://pasteboard.co/Kowd0wCMoLxW.png). The results indicate that overparameterization does not significantly degrade the performance of causal discovery and latent representation inference and, in fact, offers flexibility when there is uncertainty about the true latent dimensionality.
>
>
> **References**
>
> [1] Khemakhem, Ilyes, et al. "Variational autoencoders and nonlinear ica: A unifying framework." International conference on artificial intelligence and statistics. PMLR, 2020.

---

> > ### Comment · Reviewer_WcLB · 2025-04-04
> >
> > Thank you for the author's response. I am satisfied with the response that most of my questions are addressed.
> >
> > I will keep my score leaning towards acceptance.

---

### Official Review · Reviewer_YaAh · 2025-03-17

**Overall Recommendation:** 3

**Summary:**

The submission present SPACEY a spatio-temporal causal model which is specified over a reduced latent representation.
The authors, show that (in the continuous grid case) the model is partially identifiable, moreover the same insights should apply to the finite grid case, when the number of locations is large.
The model is implemented with non-linear SCM and non-linear pointwise plus linear mixing between spatial locations.
Inference on the defined model is performed using variational inference.

**Claims And Evidence:**

The main claims are about identifiability, and proofs are provided in the appendix (which I did not check).
Claims about superiority of the model are supported by simulation experiments.

**Essential References Not Discussed:**

yes for instance, check previous response

- Friston, Karl J., Lee Harrison, and Will Penny. "Dynamic causal modelling." Neuroimage 19.4 (2003): 1273-1302.
- Friston, Karl J., et al. "Dynamic causal modelling of COVID-19." Wellcome open research 5 (2020): 89.
- Stephan, Klaas Enno, et al. "Ten simple rules for dynamic causal modeling." Neuroimage 49.4 (2010): 3099-3109.
- Friston, Karl, Rosalyn Moran, and Anil K. Seth. "Analysing connectivity with Granger causality and dynamic causal modelling." Current opinion in neurobiology 23.2 (2013): 172-178.

**Experimental Designs Or Analyses:**

Yes I checked the validity of the simulation experiments and the setting is correct , I did not find any issue.

**Methods And Evaluation Criteria:**

Yes the proposed simulation experiments and climate examples make sense.

**Other Comments Or Suggestions:**

- check capitalization in references, e.g. Bayesian

**Other Strengths And Weaknesses:**

The paper is generally well written even if I feel the authors should make more effort to analyze their proposed method with respect to already existing methodologies,

**Questions For Authors:**

How the proposed approach relate to DCM? what are the specific differences? what are the similarities? is DCM not applicable in the simulation experiments or the climate example?

**Relation To Broader Scientific Literature:**

I think the authors lack a better discussion of the relationship between the proposed approach and classical multi-level Bayesian models.
Variational inference used in the submission is an approximation of Bayesian inference, and the specification of the SPACY model resemble a Bayesian multilevel models for spatial data. This has already been applied to a very similar setting through the so called Dynamic Causal Model (DCM).
For instance check the following references:
- Friston, Karl J., Lee Harrison, and Will Penny. "Dynamic causal modelling." Neuroimage 19.4 (2003): 1273-1302.
- Friston, Karl J., et al. "Dynamic causal modelling of COVID-19." Wellcome open research 5 (2020): 89.
- Stephan, Klaas Enno, et al. "Ten simple rules for dynamic causal modeling." Neuroimage 49.4 (2010): 3099-3109.
- Friston, Karl, Rosalyn Moran, and Anil K. Seth. "Analysing connectivity with Granger causality and dynamic causal modelling." Current opinion in neurobiology 23.2 (2013): 172-178.

DCM solves a similar problem with a very similar approach and I think a deep discussion between the similarities and the differences between the proposed approach and DCM is lacking.

**Theoretical Claims:**

Proofs are presented in the supplementary materials, I did not checked correctness.

---

> ### Author Rebuttal · Authors · 2025-04-01
>
> We thank the reviewer for their insightful feedback.
>
> We agree that both DCM and SPACY formulate the problem of inferring causal relationships between latent variables as a multi-level Bayesian model. However, they differ significantly in their assumptions and approaches.
>
> DCMs aim to infer the causal relationships of interactions in a **dynamical system** with potentially cyclic relationships. In contrast, SPACY uses the **structural causal model** [1] framework, where the causal graph is constrained to be acyclic. Moreover, DCM assumes that the parameters of the forward model (i.e. the relationship between the latent and observable variables) are known apriori. SPACY, on the other hand, infers the parameters of the forward model, as well as the correspondence between the observed variables and latent variables by fitting neural networks to the observed data.  This latent-space approach enables SPACY to explore causal relationships without relying on prior mechanistic assumptions or predefined dynamical models, making it suitable for application to high-dimensional data with unknown dynamics. Algorithmically, DCM uses the EM approach for inferring the structure, while SPACY uses Variational Inference.
>
> We will expand the related work section by including the references listed by the reviewer, and the above discussion. We will also fix the capitalization issue in the references.
>
> **References**
> [1] Pearl, Judea. "Causal inference." Causality: objectives and assessment (2010): 39-58.

---

### Decision · Program_Chairs · 2025-05-01

**Decision:**

Accept (poster)

**Comment:**

This paper proposes SPACY, a framework for causal discovery from spatiotemporal data that maps high-dimensional observations to latent variables using spatial kernels and infers their causal structure via variational inference. It offers identifiability results in a continuous spatial setting and demonstrates strong empirical performance on both synthetic and real-world data.

Reviewers found the theoretical and empirical contributions valuable but raised concerns about the strength of the modeling assumptions—specifically, that observations are generated through a nonlinear function of the product of latent series and spatial factors. While this supports identifiability, describing it as a “minimal assumption” may be misleading, as it shifts rather than weakens assumptions. Additional concerns included the realism of the spatial modeling in climate data and the limited discussion on predictive evaluation.

The authors responded constructively with new experiments and clarifications. However, it would be helpful for the paper to include a comparative summary of assumptions across prior work, and a more specific title that reflects the modeling structure. Furthermore, while the causal discovery algorithm (e.g., Rhino) used in the latent space can, in principle, be replaced, its assumptions are part of the overall modeling pipeline and should be acknowledged and discussed.

Finally, there was also some background discussion regarding the paper's title, 'Discovering Latent Structural Causal Models,' since it is well-known in the field that Structural Causal Models are not learnable from observational data, as formalized by the Causal Hierarchy Theorem (Bareinboim et al., 2022). I would recommend changing the title and/or qualifying the precise assumptions, and adding further explanation that the paper's goal is not to learn a general SCM (a provably impossible task), but rather to identify a type of equivalence class.

I recommend acceptance with slight reservation, contingent on incorporating the promised revisions and clarifying the scope and assumptions more explicitly.